



# Can integrative catchment management mitigate future water quality issues caused by climate change and socio-economic development?

Mark Honti[1], Nele Schuwirth[2], Jörg Rieckermann[2], and Christian Stamm[2]

[1]MTA-BME Water Research Group, Hungarian Academy of Sciences, Műegyetem rkp. 3, Budapest, H-1111, Hungary
[2]Eawag: Swiss Federal Institute of Aquatic Sciences and Technology, Ueberlandstrasse 133, Dübendorf, CH-8600, Switzerland

*Correspondence to:* Mark Honti
(mark.honti@gmail.com)

**Abstract.** Catchments are complex systems where water quantity, quality and the provided ecological services are determined by interacting physical, chemical, biological, economic, and social factors. The awareness of these interactions led to the prevailing catchment management paradigm of Integrated Water Resources Management. The design and evaluation of solutions for integrated water resources management requires to predict changes of local or regional water quality, which requires in-

tegrated approach for modeling too. On one hand, integrated models have to be comprehensive enough to cover the aspects relevant for management decisions, allow for mapping of global change processes – as climate change, population growth, migration, and socio-economic development – to the regional and local contexts. On the other hand, models have to be sufficiently simple and fast enough to apply proper methods of uncertainty analysis, which can consider model structure deficits and propagate errors through the chain of submodels. Here, we present an integrated catchment model satisfying both objec-

tives. The conceptual 'iWaQa' model was developed to support the integrated management of small streams. It can predict both traditional water quality parameters like nutrients and a wide set of organic micropollutants originating from plant and material protection products. Due to the model's simplicity, it allows for a full, propagative analysis of predictive uncertainty, including certain structural and input errors. The usefulness of the model is demonstrated by predicting future water quality in a small catchment with mixed land use in the Swiss Plateau. The focus of our study is the change of water quality over the

next decades driven by climate change, population growth or decline, socio-economic development and the implementation of management strategies for improving water quality. Our results indicate that input and model structure uncertainties are the most influential factors on certain water quality parameters and in these cases the uncertainty of modeling is already very high for the present conditions. Nevertheless, a proper quantification of today's uncertainty can make the management fairly robust for the foreseen range of possible evolution into the next decades. With a time-horizon of 2050, it seems that human land

use and management decisions have a larger influence on water quality than climate change. However, the analysis of single climate model chains indicates that the importance of climate grows when a certain climate prediction is considered instead of the ensemble forecast.





# 1 Introduction

Catchments are complex systems where water quantity, quality and the ecological services provided are determined by interacting physical, chemical, biological, economic, and social factors. The awareness of these interactions led to the prevailing catchment management paradigm of Integrated Water Resources Management (IWRM) (GWP, 2000). IWRM relies on the
conviction that none of these factors can be altered without influencing water quantity and quality and hence all aspects need to be considered in sustainable water management and when evaluating different management alternatives. Unfortunately, it is not trivial to assess the influence of many of these factors. While large-scale processes like climate change or socio-economic development propel global change, understanding the effects of global change on water resources requires focusing on the regional or even local context. Improving water quality at the outlet of a catchment requires a reduction of relevant point or
non-point sources of pollution within the catchment, which depend on land-use and infrastructure. Therefore, the effectiveness of a particular management strategy may strongly differ between catchments. Some water quality management strategies require long-term investments, e.g. improving the wastewater infrastructure. Hence scenario analyses about possible future developments and their influence on the effectiveness of water quality management strategies are crucial for decision making in this field. Just like any intervention into a complex system, water quality management can also have undesired side-effects.
A blind application of management recipes can actually have a partly undesired outcome. For example, the reduction of suspended solids load increases water transparency, which, given enough residence time in large rivers like the Danube can lead to advanced eutrophication (ICPDR, 1999). This could have been forecasted by an integrated modelling approach, but side-effects can also come from outside the simulation domain of catchment models: the revitalisation of a section of the Aare River in Switzerland has improved recreational attraction, which caused an unexpected increase in disturbance and waste load (Witschi
and Käufeler, 2014).

To reliably address the effects of Global Change on regional or local water resources, a potentially useful model must include all major factors and transport processes that control water quantity and quality at the catchment scale. For IWRM, this is particularly challenging because most managed catchments contain both rural (natural and agricultural) landscapes and settlements in varying proportions. This holds especially true for densely populated areas like the Swiss Plateau, where we
apply our approach. Accordingly, an ideal model has to have sufficient coverage of both rural and urban processes.

In this regard, we reviewed existing and potentially suitable tools and found that all of them had deficits in this aspect. Specifically, the majority of popular models seems to be focusing on either rural (AnnAGNPS [Bingner and Theurer, 2001], ANSWERS [Bouraoui et al., 2002], CREAMS [Knisel, 1980], GWLF [Haith and Shoemaker, 1987]) or urban (SWMM [Metcalf and Eddy Inc. et al., 1971]) areas. The remaining two universal models (HSPF [Donigian et al., 1995], SWAT [Arnold
et al., 1995]) lack sufficient detail in the descriptions of urban water infrastructure. For example, they do not include combined sewer overflows (CSO), which often play an important role for water quality as emitters of pesticides into streams in urban areas (Wittmer et al., 2010) and the design of urban water infrastructure alternatives. Important processes of rural catchments are often lacking too. Some models calculate the emission and transport of nutrients only (ANSWERS, GWLF, PhosFate [Kovacs et al., 2008]), while others include routines to simulate pesticide loads and concentrations as well (AnnAGNPS, CREAMS,





HSPF, SWAT). In summary, none of these readily available models simulate all important pollutant classes and their sources for typical subjects of local water quality management: small catchments covering tens of km2 and having both settlements and agricultural areas. A detailed comparison of these models is presented in section S1 of the Supporting Information (SI).

The preceding analysis of model comprehensiveness might suggest the use of more complex models in order to simulate as many processes as possible and therefore restricting the number of phenomena not covered by the model. There is indeed intensive research towards model frameworks that combine specialised models into an integrated system. Certain model frameworks address certain micropollutants (see Vezzaro et al. (2014) for a case with urban water systems; Bloodworth et al. (2015) for drinking water risk assessment) and some even cover certain economical aspects (Boehlert et al. 2015). There is a high expectation towards integrating more modules, data and knowledge into ever growing model systems (Holzkämper et al., 2012; Hipsey et al., 2015; Salvadore et al., 2015). However, we argue that models used for management purposes should not only represent the relevant aspects of the managed system but also allow for a realistic and technically feasible assessment of the uncertainties linked to model predictions. The statistically valid estimation of prediction uncertainty is essential to make robust decisions about management options, but can be hardly addressed in large frameworks.

Catchment models have a complex hierarchical structure ranging from basic variables that affect many others, such as hydrological flow components and water temperature, to endpoints, on which no other variable depends, such as specific pollutant concentrations. Therefore, a proper uncertainty assessment involves a mechanistic propagation of uncertainty through this hierarchy. Since such models are usually strongly non-linear, numerical methods are needed to propagate parameter uncertainty and for model calibration. Bayesian inference allows taking into account prior knowledge about parameters as well as data on output variables to estimate the posterior parameter distribution and model output uncertainty (Gamerman, 1997). However, this requires tens to hundreds of thousands of model runs. The quantified parameter uncertainty is then transferred to the predictions by again thousands of predictive runs on the combination of possible model and error parameter values. Thus, the creation of such probabilistic predictions is extremely resource-intensive, which practically excludes all computationally expensive models from such an analysis.

Hence, there is a need for water quality predictions, which are comprehensive enough to cover the aspects relevant for management decisions, allow for coupling of global change processes with the regional and local contexts, but are based on sufficiently simple models, which allow for proper uncertainty analysis that propagates uncertainty through the entire model chain. Accordingly, the objectives of this paper are threefold.

First, we introduce a conceptual catchment model (the iWaQa model) that was developed for small streams with IWRM-specific objectives in mind (simple, consistent and comprehensive in terms of both pollutants and pollution sources). Due to the model's simplicity, it is possible to estimate parameters from data and to perform a nonlinear error propagation with Monte Carlo techniques in a "total uncertainty analysis" framework, which overcomes the major limitations mentioned above.

Second, we investigate how water quality in catchments may develop over the next decades until 2050, considering the impacts of climate change, socio-economic development and the implementation of management strategies to improve water quality. To this aim, we apply the model to a small catchment with mixed land use in the Swiss Plateau, where a very good data coverage and a wealth of field observations are both available to condition the model.





Finally, we discuss how the insights gained from the application of the model to the case study can contribute to improved management of water quality under global change in general.

The main novelties of this paper are the following: (i) The integrated analysis of climate change, socio-economic development and management, covering boundary conditions over a wide spectrum of sources and scales; (ii) a comprehensive, "total uncertainty analysis" along the entire integrated modelling procedure. This not only considers uncertainty in inputs, model structure, model parameters, and output observations, but also accounts for uncertainty from our current lack of knowledge regarding important boundary conditions by ensemble forecasts; (iii) Modelling the dynamic concentration of important organic micropollutants, such as plant and material protection products, based on observed data; (iv) Carrying out the case study in a mixed-landuse catchment, which receives important pollutant contributions from both agriculture and urban settlements.

## 2 Methods

### 2.1 Hydrology

We use the modified LogSPM conceptual model (original: Kuczera et al. [2006], modified version: Honti et al. [2013; 2014]) to simulate stream discharge at the catchment outlet due to its simplicity and acceptable performance of its varieties in several catchments (Kuczera et al., 2006; Reichert and Mieleitner, 2009; Honti et al., 2013). This conceptual model describes runoff formation by assuming a function that unambiguously maps between average soil moisture content and saturated fraction of catchment surface. The model produces both total streamflow and individual flow components, namely baseflow, subsurface flow and surface runoff. A detailed description of the model and calibration procedure can be found in Honti et al. (2013, 2014).

### 2.2 Separation of relevant flow components

The hydrological needs of a conceptual model aiming to simulate pollutant transport are usually different from the needs of a hydrological catchment model. Due to the specific transport pathways of pollutants one needs to consider water fluxes that are extremely important for the propagation of a given pollutant family but may not be of any particular importance for streamflow on the catchment or sub-catchment scale. This means that – building on the modelled flow components – we have to use a more elaborate flow routing scheme that includes all important pollutant transport pathways but remains easily derivable from the catchment-scale flow components (baseflow, subsurface flow, runoff).

Since the majority of pollutants originate from settlements and agricultural areas, these subsystems deserve a more detailed hydrological description (Fig. 1). From a hydrological perspective the behaviour of a rural catchment is dominated by water storage in different soil horizons. Urban areas need to be relatively large to exert a detectable effect on the flow regime on aggregated time-scales (e.g. days). Consequently, conceptual rainfall-runoff models typically lack a detailed urban hydrology module that would represent the different building blocks of the urban water infrastructure, as they would be anyway unidentifiable in the output.



We divide catchment-scale flow components between more detailed flow paths according to simple linear partitioning rules (section S2 in the SI). Partitioning parameters are calibrated by fitting the model to observed concentrations of traditional inorganic pollutants like nutrients or major ions, such as sulphate.

### 2.2.1 Water temperature

Water temperature is simulated in a semi-distributed empirical way (section S2 in SI). We used the steady state solution of the full process-based stream temperature model of Meier et al. (2003) to estimate daily mean equilibrium temperatures, which is the water temperature at which the net heat flux through the air-water interface is zero under the given meteorological conditions (Edinger et al., 1968). The bulk heat exchange through the air-water interface is then proportional to the difference between the actual and equilibrium temperature (Edinger et al., 1968) and thus the temporal evolution of temperature follows first-order kinetics. For the description of heat transport, we adopted the semi-Lagrangian plug flow approach by Yearsley (2012) combined with a discharge-weighed averaging of the travel time to depth ratios. This was a good approximation of the full solution of the plug flow heat transport problem in small catchments (Tendall et al. *in prep*).

### 2.2.2 Water quality

Our model is a simplified stream quality model because concentrations are calculated from stream loads and discharge. In-stream processes are neglected, which is an acceptable simplification for small streams with limited residence time. The detailed description of water quality calculations is presented in section S2 of the SI).

We use the separation of total flow into different pollution transport pathways to calculate nutrient concentrations by associating each pathway with a typical concentration and mixing these components proportionally in the stream. Mixing is linear, except for pH (section S2 in SI). Elimination or transformation of pollutants inside the wastewater treatment plant (WWTP) is described by first-order models based on long-term mean removal/transformation efficiency and the relative hydraulic residence time inside the WWTP.

Out of the large diversity of organic micropollutants, we focus here on pesticides (including biocides for protecting materials and plant protection products [PPPs]). They consist of biologically active ingredients, occur frequently in Swiss surface waters (Moschet et al., 2014) and have both agricultural and urban sources (Wittmer et al., 2010). These compounds are treated in a similar way after being mobilised as traditional pollutants: fluxes from various sources are gathered by the various converging flow components and in-stream concentrations are calculated by division by stream discharge. For biocides an additional assumption is that the source stock is constant and proportional to the application area in the catchment. PPPs are assigned a dynamically changing stock that considers application amounts and periods and a unified dissipation process that accounts for all loss processes other than export to the stream network, such as in-situ biotransformation, photodegradation, ageing, etc.



### 2.2.3 Uncertainty assessment

The uncertainty of model and error parameters was assessed with Bayesian uncertainty assessment separately for each sub-model. The likelihood for total discharge was determined using a composite error model containing a heteroscedastic autoregressive process, the so-called "model bias" and independent Gaussian observation errors (Honti et al., 2013). Conceptually, the Gaussian errors can be interpreted as random observation errors and the bias covers the remaining systematic discrepancy between the model and the data.

For water temperature data and traditional pollutants a simpler error model (independent normal errors) was applied satisfactorily. The same error model was used for micropollutants, but a Box-Cox transformation with an exponent of 0.3 was applied to both the observed and modelled time-series to account for heteroscedasticity originating from the high variability of these data.

Posterior parameter distributions were estimated with Markov chain Monte Carlo sampling using the traditional Metropolis algorithm (Gamerman, 1997). Chain length varied between 100,000 and 500,000 iterations depending on convergence between 3 independent chains. Predictive model output uncertainty was estimated by propagating the predictive uncertainty of all variables through the submodel hierarchy. A random sample of 1000 iterations was used for uncertainty propagation and the estimation of 95% uncertainty intervals.

## 3 Case study

### 3.1 Location

The Mönchaltorfer Aa catchment (NE Switzerland, area=43 km$^2$) is a typical representative of catchments with mixed landuse on the densely populated Swiss Plateau. Landuse is dominated by intensive agriculture (57%); other important categories are forests (15%) and settlements (11%). Presently, five villages are home to 24 000 inhabitants. However, the area is just 20 km away from the booming metropolitan centre of Zürich; therefore intensive suburbanisation is likely to take place in the close future if the economy of Switzerland keeps growing dynamically (see socio-economic scenarios below). Urban areas are gaining ground against agriculture in most parts of the Swiss Plateau (Lanz et al., 2014) due to economic pressure from elevated property prices. These potentially dynamic socio-economic conditions and the good data coverage (Wittmer et al., 2010) made the Mönchaltorfer Aa catchment an interesting subject for our analysis.

### 3.2 Water quantity and quality variables

The water quantity and water quality variables modelled in the case study are summarised in Tab. 1. The model calculated daily mean values for all variables. Micropollutant concentrations were aggregated to weekly means during calibration to be comparable to the similarly aggregated observations. The selection of the micropollutants considered in this study was largely driven by the availability of measured data.



### 3.3 Model calibration

Catchment hydrological parameters were calibrated based on 10 years of daily discharge and precipitation data (2000-2009) measured at the catchment outlet (AWEL, 2010); see Honti et al. (2014) for a detailed description. The site-specific exposure parameters for atmospheric heat exchange (Tendall et al. in prep) were calibrated based on 3 years of daily mean water temper-

atures (2004-2006, [AWEL, 2010]). Discharge time-series were decomposed into fast, medium and slow response components by recursive digital filtering (Eckhardt, 2005; Rimmer and Hartmann, 2014) to provide estimates of runoff, subsurface flow and base flow for the measured discharge data as well.

Parameters controlling urban hydrological pathways and traditional pollutant behaviour were calibrated based on a 3 years' daily concentrations of Cl, $SO_4$, DO, $NO_3$, $NH_4$, $PO_4$, and TP at the catchment outlet (2004-2006, [AWEL, 2010]). These water

quality parameters contain good tracers of sewage (Cl in summer only because of its use in winter as de-icing agent on roads; $SO_4$ all year), a good indicator of WWTP treatment efficiency ($NO_3$) and a good indicator of fresh organic matter emissions ($NH_4$). By utilising the many-faceted behaviour of these pollutants we could calibrate the partitioning parameters that distribute catchment-scale flow components among the modelled pollutant transport pathways. For micropollutants we used the already calibrated transport pathways and calibrated the compound-specific application, persistence, and export parameters separately.

### 3.4 Boundary conditions for prediction: scenarios of climatic change and socio-economic development

The time-horizon for our predictions was the year 2050, so we carried out simulations for the period 2035–2064. For future climate we used stochastically downscaled data from 10 ENSEMBLES GCM-RCM model chains for the IPCC A1B emission scenario. We did not consider additional emission scenarios because they are practically the same during our prediction period. Details about the stochastic downscaling procedure can be found in Honti et al. (2014).

We used four scenarios for socio-economic development (Tab. 2) based on a workshop with local stakeholders and predictions about the regional population driven by domestic and international migration (Lienert et al., 2014). Three out of the four scenarios covered moderate growth, stagnation, and moderate decline in population, with moderately growing and stagnant urban area, respectively. In contrast, the fourth scenario described 'exploding growth' in population with a simultaneous radical increase in settlement area.

'Moderate growth' was envisioned as an environmentally friendly way of development, supported by the reportedly high environmental attitude of the Swiss population (Franzen and Vogl, 2013). Moreover, regardless of the attitude of the local society, moderate growth can also represent a gradual landscape development typical to Central Europe in the last century (Niedertscheider et al., 2014). The rather extreme 'exploding growth' scenario described an intensive urbanisation in about half of the total catchment area with a 7-fold increase in population. While this may seem excessive in a developed country

like Switzerland, it was still considered feasible by local stakeholders, probably due to the several precedents of a comparable level of regional urban development in the last decade in the large metropolitan area of Zürich (Lienert et al., 2014).





## 3.5 Management alternatives

As integrated management was in the focus of our study, we investigated management alternatives that could improve present and future water quality (Tab. 3). Since societal adaptation to climate change covers various spatial, temporal and organisational scales (Adger et al., 2005), we investigated management actions on both local and wider political scales. An example for local actions would be the adaptation of the urban water infrastructure, whereas an example for management on a wider political scale could be the regulation or complete ban of the use of certain PPPs. This would have to be implemented through cantonal (autonomous administrative units of Switzerland) or federal regulations and policies.

Specifically, we investigated the impact of the following management alternatives: i) the regulation of material protection to reduce biocide emissions into streams, ii) adaptation of urban water infrastructure to reduce urban runoff and storm water emissions, iii) the upgrading of WWTPs to enhance the removal of nutrients and micropollutants (Eggen et al., 2014, see e.g.), and iv) reducing diffuse pollution from agricultural areas. For the sake of comparison, we also predicted future water quality given the 'current practice' and the 'total management' alternative, the latter combining all management measures and thus providing information regarding the best achievable outcome. The latter two alternatives delineate the degree of freedom of management under given socio-economic and climatic conditions. The translation of management alternatives into quantified changes in catchment properties, emissions, and other boundary conditions is presented in section S4 of the SI.

## 3.6 Comparison of predictions under different boundary conditions

Our model predictions for water quality were driven by different realisations of the 10 stochastic climate processes belonging to the 10 GCM-RCM model chains (Honti et al., 2014). As opposed to a 'delta change' approach, our dynamic weather generation approach meant that we had independent realisations of weather for each model chain, so – except for a seasonal correlation –there wasn't any connection between the alternative values of modelled discharge for a given day in the future. Thus, the change in the statistical properties of discharge had to be analysed instead of the differences between the time-series (Honti et al., 2014). The same argument holds also for the time series of water quality parameters. Therefore, to summarise changes, we analysed the relative difference in the extreme high (97.5%), median (50%), and extreme low (2.5%) quantiles from the simulated series for each variable.

## 4 Results

### 4.1 Model calibration and main sources of pollution

Daily discharge, mean water temperature and certain traditional water quality parameters (Cl in summer, $SO_4$, DO, and $NO_3$) could be calibrated well (Fig. 2, Nash-Sutcliffe index NS = 0.75 [$NO_3$] – 0.86 [$SO_4$]). Pollutants bound to rare, very intense precipitation events through stormwater overflows and soil erosion were calibrated with substantial uncertainty ($NH_4$, $PO_4$). TP was found to be the most problematic traditional pollutant due to its connection to intense rain events that cause strong yet seasonally variable particulate P peaks via erosion and some continuously high-TP periods in the observation data in 2006



stemming from sources which could not be identified with certainty (a plausible explanation could be in-stream construction works in the upstream network).

Most PPPs show a clear seasonal pattern as the largest stream concentrations occur during the first few storm events after the application (Leu et al., 2004, 2005). This expressed seasonality of concentrations helped the model to attain a reasonable

overall performance, despite the significant uncertainty of individual event concentrations (Fig. 3). The situation was different for biocides. Due to their intentional persistence, biocide stocks were less variable throughout the year, so the uncertainty of the numerous individual peaks resulted in an almost uninterrupted high concentration uncertainty during all rain events (Fig. 3). We present the results for the different uncertainty assessment approaches by study site.

## 4.2  Future water quality

The combination of four socio-economic scenarios, the ten GCM-RCM model chains and ten management alternatives resulted in 400 combinations of boundary conditions for our single simulation site. Although we modelled all combinations, for the sake of clarity we first compare the effect of the three main classes of boundary conditions (climate, socio-economic development, and management) separately.

Predictive uncertainty was often very high for the boundary conditions of the calibration period (Fig. 4). While variables

like water temperature and most nutrients were simulated very well, predictions for pollutants like biocides or PPPs were much more uncertain. We identified model structure deficits and input uncertainties (e.g., how much of a given biocide is actually applied in a catchment and where) as the major sources of uncertainty.

## 4.3  Climate change

The possible effect of climate change on water quality strongly depends on whether all climate models are pooled or whether

one investigates single climate models individually (Fig. 4, section S5 in the SI). If all climate models are pooled (reflecting the overall current uncertainty about climate change) the climate-driven changes were generally negligible compared to prediction uncertainty for micropollutants, indicated by the similar position of black and red bars in the upper panel of Fig. 4. The reasons for the large uncertainty are the strong deviations among the climate models regarding future precipitation patterns. Contrary to the increasing air temperature, there was no qualitative consensus among the ten different GCM-RCM chains in the future

evolution regarding precipitation. This caused a corresponding divergence in pollution dynamics (Tab. 4). However, because only one climate change path will develop, the analysis of single model chains can give insight how strong the climate signal may affect water quality in the future. Water quality may change substantially depending on the actual future climate (Fig. 4, comparison of the upper and lower panel). With the ETHZ HadCM3Q0 CLM model chain, runoff was predicted to decrease in a significantly drier future climate, which also reduced the wash-off of PPPs and biocides from fields and buildings, respectively.

A wetter climate – e.g., the SMHI BCM-RCA model chain – is predicted to do the opposite, i.e. increasing the loss rates of many micropollutants (see Tab. 4).





## 4.4 Socio-economic development

Socio-economic development was found to be a key determinant of water quality, which was especially impaired by the expansion of urban settlements at the costs of the existing agricultural area. However, changes were again negligible compared to the uncertainty and present variability, except for the "Exploding growth" scenario (Tab. 5). As expected, the "Decline"
scenario caused no change, except for a slight decrease in water temperature due to the decreased discharge of WWTP effluents. In similar fashion, the "Moderate growth" scenario only resulted in a small increase of biocide concentrations, due to the 5% expansion of settlements. Here a counter-decrease in agricultural pesticides could not be observed, because the urban expansion extends only over 1% of the total arable land.

The huge changes envisioned in the "Exploding growth" scenario lead to predictions where changes in the mean exceeded
both prediction uncertainty and intrinsic variability. Changes in discharge led to an increase in high flows and a decrease in low flows due to the immense new paved areas. Water temperature rose due to the 8-fold increase of WWTP effluents. PPPs decreased due to the reduction of arable land, biocides increased due to the growing number and size of buildings.

## 4.5 Management

As mentioned above, the effects of management alternatives are case study specific, because they depend on the dominant
sources of pollution in the catchment. In the Mönchaltorfer Aa, most management alternatives had significant impact on their water quality targets according to our model predictions (Tab. 6), except the StoreVol alternative. The failure of StoreVol is partially due to our model's structure and partially due to real phenomena because increasing the storage volume leads to effects with contrasting influences on pollutant discharge. On the one hand, the increased buffer volumes in the drainage network decreased overflow volumes into streams. On the other hand, this increased the hydraulic load at the WWTPs and
caused a decrease in modelled WWTP removal efficiency, which was comparable to the pollution prevented at the CSOs. The overall modelled effect was that there was no noticeable decrease in pollutant loads. This outcome can be regarded as an estimate of the real, only slightly reduced pollutant loads that could be achieved in reality by optimising WWTP operation with regard to hydraulic stress.

For the remaining management alternatives, the only side-effect with regard to micropollutants appeared for certain biocides:
efforts to reduce urban runoff (PermPave, RetRain) on one hand decreased flood peaks, but on the other they reduced the dilution capacity for biocide pulses. Overall, this resulted in elevated biocide peak concentrations, when the biocide's sources were outside of runoff retention zones.

## 4.6 Robustness of changes, management potential

The joint analysis of all predictive combinations allowed us to find out general change patterns and examine the potential effi-
ciency of management measures under future conditions. Pooling the predictions of all socio economic and climate scenarios into a single set blurred the often substantial differences between predictions based on individual boundary conditions (for example, the effect of individual GCM-RCM chains in Fig. 4). As a result, small differences disappeared and others, which



were probably less significant but showed up in many combinations, were revealed (Figs. 5-7). The following findings were obtained for a 'fully uncertain' future, which contained the full set of all predictions under all socio-economic scenarios and climate change predictions around 2050 (where for example "exploding growth" is just as likely as "decline").

Discharge is not expected to change significantly in the fully uncertain future (Fig. 5). In the present climate the Sta-
tusQuo+All combination had the lowest ($12.3°C$) and the Exploding Growth+CurrPrac setup had the highest ($13.7°C$) mod-
elled annual mean water temperature. Mean water temperatures show a clear increase in the future by $+1.01$ to $+1.16$ $°C$ due to the warming climate. This seems to be difficult to mitigate substantially by increasing riparian shading through the restocking of riparian buffer zones. Without a change in management practice, traditional pollutant concentrations will remain similar to their present values. $NO_3$, Cl and $PO_4$ can be managed to some degree by the combined management alternative, with
$NO_3$ showing the largest decrease. Other pollutants ($NH_4$, $SO_4$) remain similar, regardless of management. This indicates that, with the exception of $NO_3$, traditional water quality variables can only be managed efficiently when the uncertainty of future boundary conditions is reduced by examining specific scenarios: not even the "All" alternative is effective under fully uncertain future boundary conditions.

There are no differences due to climatic and socio-economic changes for PPPs (Fig. 6, overlap of the grey area and black
line). The efficiency of management fundamentally depends on whether a compound is affected by a future ban (legislative ban or abandoning the cultivation method it belongs to: atrazine, isoproturon, metolachlor, 2,4-D, diazinon, MCPA, terbuthylazin) or not (DEET and pirimicarb). Biocides showed a uniform expected increase for the future due to the chance of extensive urbanisation (Fig. 7). By the combined management options this increase could be reduced to present levels for mecoprop or even further for the two compounds affected by a ban in "BanBioc" (diuron and terbutryn).

In summary, the iWaQa model could predict river water quality in our small case study catchment for traditional physical-
chemical variables, and organic micropollutants under future boundary conditions. We found that future prediction uncertain-
ties were generally large. Unfortunately, despite of the excellent information base, the uncertainties of present water quality predictions are already excessive for certain variables. Our results suggest that input and model structure uncertainties are often the most influential factors of future water quality.

# 5 Discussion

Despite the large prediction uncertainty, we think that our approach can effectively support river water quality management. To better interpret the obtained results, we would like to discuss in this section the following points: i) uncertainty, natural vari-
ability and change signal, ii) relative importance of boundary conditions, iii) evaluation of the transferability of the approach, and iv) whether models should be used at all for such long-term predictions.

## 5.1 Uncertainty, natural variability and change signal

Understanding the sources of predictive uncertainty regarding water quality is essential from a scientific as well as from a practical point of view. It can guide further research and management decisions. The model concepts and its implementation





presented in this paper can be used to quantify different possible sources of predictive uncertainty like climate change, socio-economic development or model and input uncertainty. We found that for some water quality parameters predicted future changes are much smaller than the observed quantile variability and predictive uncertainty together. This holds mainly for quantities that are difficult to predict under current conditions due to model structure deficits and input uncertainty (e.g.,

certain PPPs and biocides). While observed variability is outside a modeller's scope, the quantified model uncertainty could be most probably substantially reduced for certain compound groups, most notably the biocides. A more detailed identification of sources and application patterns could improve the fit between model simulations and observations for the past, which is a necessary first step towards reducing predictive uncertainty.

### 5.2   Relative importance of boundary conditions

We ranked boundary condition categories by the effective impact on pollutant concentrations caused by them. For pollutants banned in a certain management alternative, management was the most powerful factor, followed by socio-economic development and climate change at the last place. For compounds not to be banned in a certain management alternative, socio-economic development swapped places with management. Traditional pollutants like nutrients cannot be banned, so they all belonged to the second class.

This ranking highlights that drastic management measures (decided predominantly on national or regional level) can potentially mitigate the effects of almost any socio-economic or climatic change from the exposure aspect. However, this does not apply to all variables and technical possibility does not mean that such management solutions necessarily become politically or financially affordable and sustainable in the future. Because the most drastic and therefore the most efficient measures, such as a ban for certain chemicals, are predominantly decided on a national or regional level, these options involve more stakeholders,

influence factors and slower implementation compared to decisions to be taken locally.

One reason behind the relatively small importance of climate change is that our predictive time horizon positioned around 2050 is relatively close to the reference period, which meant that changes compared to the present climate were limited. A projection to the year 2100 could alter this outcome dramatically. Obviously, a few percent changes of climatic parameters could not catch up with the often extreme development and management scenarios. Thus, it should be kept in mind that the

above importance ranking is conditional on the boundary conditions of our study. That said, we would like to emphasize that, although future water quality seems to be comparably insensitive to the impact of the ensemble climate predictions, there were often significant differences between the predictions based on individual model chains. Water quality differences between the driest and wettest model chains were comparable to changes caused by the limited development scenarios ("Status quo", "Moderate Growth", "Decline"). However, aggregation into an ensemble forecast converted these differences to uncertainty.

Given the low relative importance of the climatic boundary conditions for most water quality parameters in our case study, the computational burden caused by the proliferation of combinations of boundary conditions could be reduced by neglecting climate change for all variables except discharge and water temperature. This would have reduced the number of simulations by a factor of 10. However, it is arguably difficult to make such a decision a priori.



## 5.3 Transferability of the approach

The prediction of local water quality requires skilful choice between available mathematical models. The emphasis to be put on processes taking place in the catchment or in the stream depends on the size of the stream network. In small catchments with short water residence times of only a few hours, in-stream transformation can usually be neglected. Compounds that are not stable in this timeframe have already undergone degradation upstream, e.g. in the sewers. In contrast, for large rivers that are much bigger than their individual tributaries, travel time gets comparable to residence times in small lakes and therefore in-stream processes can completely suppress the influence from the adjacent catchment (Istvánovics et al., 2014).

Not surprisingly, our approach faces the same challenges as any other integrated catchment modelling approach:

1. The relatively simple iWaQa model relies entirely on calibration. Therefore, it requires a lot of data for the past, which may not exist everywhere. More complicated models tend to hide the necessity of calibration, but they can't get rid of it completely.

2. Even though the iWaQa model is conceptual and rather simple, it considers many catchment processes, and thus requires a detailed definition of scenarios for future climate, socio-economic development and management alternatives. This would be even worse for a complicated model demanding more input data, like detailed land use or infrastructure maps.

3. Despite the integrated approach and calibration, the model still cannot guarantee precise forecasts due to existing uncertainty and lack of knowledge.

4. Uncertainty due to unknown mechanisms not included in the calibration data or unintended side-effects illustrated in the introduction cannot be quantified. We will discuss this further in the next section.

## 5.4 Should models be used for long-term predictions at all?

The typical simplifications in catchment models mean that they require at least partial calibration and as such they are intrinsically bound to the system they were calibrated to. Calibrated models emphasise processes that were important in the calibration dataset, and suppress others that did not significantly influence the observed behaviour. Model validation on independent data could in principle reveal if the model retained enough generality to predict "never seen before" events, but it is rare that validation data contain significantly different input and boundary conditions than the calibration dataset and the model keeps its predictive performance. Thus, there is indeed a contradiction between such an indirect conditioning of models to observed phenomena and the practical need to extrapolate with the model, i.e. predict environmental changes under potentially heavily changing boundary conditions. Unfortunately, this makes predictions by environmental models very uncertain.

Contrary to the uncertainty assessable from the mismatch between model simulations and historic observations, the potential inability of a model to work properly under never experienced boundary conditions is impossible to quantify. However, to support decision making about long-term investments in water quality improvements today, we have to rely on the best available knowledge and we need to quantify remaining sources of uncertainty to the degree possible. At best, we must be always aware that this might still be a conservative estimate and should be communicated openly.



In addition, high forecast uncertainty does not necessarily preclude efficient decision support. As shown by Reichert and Borsuk (2005), although absolute water quality changes might be very uncertain, relative differences between different alternatives are often robust and often lead to a stable ranking of management alternatives. In a similar fashion, for our study we can at least detect whether we can expect a significant positive effect from different management strategies on traditional water

quality parameters and micropollutants. This is already important information to decide which investments are justifiable under the current state of knowledge. In summary, we conclude that models shall be used also for long-term predictions because i) they can coherently summarise our current understanding and ii) can often deliver a stable ranking for a set of for water quality management alternatives, despite large absolute prediction uncertainties. However, one has to be aware that the quantifiable uncertainty yields only a lower limit of the real uncertainty, and a model cannot think "out of it's box", which makes it is

impossible to assess the consequences of certain side-effects by modelling.

## 6   Conclusions

The objectives of this paper were to integrate the current state of knowledge to analyse how water quality of catchments may develop over the next decades under the influence of climate change, socio-economic development and the implementation of different management strategies to improve water quality and to develop a model structure that makes such a comprehensive

analysis possible. The results demonstrate the usefulness of such a broad approach that incorporates the major (quantifiable) sources of uncertainty by propagating all relevant sources of uncertainty through an integrated model framework. Our analysis for the Mönchaltorfer Aa catchment in Switzerland reveals different major sources of uncertainty for individual water quality variables: for some variables such as biocides input uncertainty is dominant. These are difficult to predict even under current conditions just because accurate usage data are lacking. Interestingly, lack of input data not only limits model testing and im-

provement, but most probably increases uncertainty from model structure deficits too. Nevertheless, climate change increased prediction uncertainty for biocides, while it did not for PPPs (see Fig. 7).

From our results we can derive definite recommendations for practical water management too: Water quality analysts should always assess the uncertainty of their model simulations, not only when planning for the far future. Given the complexity of the catchment systems, some water quality parameters cannot be hindcasted with high confidence. This can be due to input as well

as model structure uncertainty. The positive aspect of this situation is that a proper accounting of this uncertainty today will make the management fairly robust for the foreseen range of possible evolution into the next decades. However, the analysis of single climate model chains also indicates that the importance of climate grows when model chains are considered individually. As a practical consequence, one should carefully follow how climate change unfolds and how it changes water quality.

Despite all uncertainties, the results clearly show that the direct human influence via the local, regional and national scales

exerts the largest effect on water quality. At least for the time horizon considered in this study (2050), land use and management options to control water quality seem to dominate over climate change effects. This provides a chance and the corresponding responsibility for active catchment management to achieve a good water quality status in the future.





*Acknowledgements.* The iWaQa project is financed by the Swiss National Science Foundation (National Research Program #61 on Sustainable Water Management, Grant #406140-125866) and the Swiss federal Office for the Environment. The ENSEMBLES data used in this work was funded by the EU FP6 Integrated Project ENSEMBLES (Contract number 505539) whose support is gratefully acknowledged.





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





**Table 1.** Water quantity and quality variables modelled in the case study

| Category | Notation | Description |
|---|---|---|
| Hydrology | Q | Discharge |
| Thermal regime | T | Water temperature |
| Tracer ions | Cl | Chloride concentration |
| | SO4 | Sulphate concentration |
| Oxygen and pH | DO | Dissolved oxygen concentration |
| | pH | pH |
| Nutrients | NO3 | Nitrate concentration |
| | NH4 | Ammonium concentration |
| | PO4 | Phosphate concentration |
| | TP | Total phosphorus concentration |
| Plant protection products (PPPs) | Atrazine | Atrazine concentration[*] |
| | Isoproturon | Isoproturon concentration |
| | Metolachlor | Metolachlor concentration |
| | 2,4-D | 2,4-dichlorophenoxyacetic acid (2,4-D) concentration |
| | DEET | N,N-diethyl-meta-toluamide (DEET) concentration |
| | Diazinon | Diazinon concentration[*] |
| | MCPA | 2-methyl-4-chlorophenoxyacetic acid (MCPA) concentration |
| | Pirimicarb | Pirimicarb concentration |
| | Terbuthylazine | Terbuthylazine concentration |
| Biocides | Diuron | Diuron concentration[†] |
| | Mecoprop | Methylchlorophenoxypropionic acid (MCPP) concentration |
| | Terbutryn | Terbutryn concentration |

[*] Now banned in Switzerland, kept as a proxy for future substitute compounds; valuable model compound given long time series in water bodies. [†] Also used as a PPP

**Table 2.** Socio-economic development scenarios for 2050

| Development Scenario | Content |
|---|---|
| Status quo | Urban area and population do not change |
| Moderate growth | 20% growth in population, 5% growth in urban area |
| Exploding growth | 730% growth in population 300% growth in urban area |
| Decline | 20% decline in population, no change in urban area |





**Table 3.** Management alternatives

| Category | Name | Content | Rationale |
|---|---|---|---|
| Current practice | CurrPrac | Current practice | (included for comparison) |
| Material Protection | BanBioc | Banning application of biocides on façades | Reduce biocide loads |
| Urban Water Infrastructure | StoreVol | Increasing storage volumes in urban drainage systems | Reduce stormwater emissions |
| | PermPave | Increasing proportion of permeable pavements | Reduce urban runoff |
| | RetRain | Retention of rainwater from roofs | Reduce urban runoff |
| End of Pipe | WWTP | Enhancing WWTP treatment efficiency | Reduce point source loads |
| Agriculture | OrgFarm | Exclusively organic farming | Eliminate agricultural pesticides |
| | BufZone | Reconstruction of riparian buffer zones | Less erosion, more shading |
| | NatPark | Nature Park | Eliminate intensive agriculture and pesticides |
| Total Management | All | All measures combined | Best available management |



**Table 4.** Signs of change for future physical parameters and micropollutant concentrations considering climate change only [assuming Status quo+CurrPrac], derived from a figure similar to Fig. 4. +: general increase (none of Q2.5%, Q50%, and Q97.5% decreased; red), −: general decrease (none of Q2.5%, Q50%, and Q97.5% increased; blue), ±: quantile-specific response (at least a pair out of Q2.5%, Q50%, and Q97.5% changed in different directions; purple), 0: no change (grey)

| Variable | Ensemble future climate (10 GCM-RCM model chains) | ETHZ HadCM3Q0-CLM (dry) | SMHI BCM-RCA (wet) |
|---|---|---|---|
| discharge | ± | − | + |
| water temperature | + | + | + |
| atrazin | 0 | 0 | 0 |
| isoproturon | + | − | + |
| metolachlor | − | − | 0 |
| 2,4-D | + | − | + |
| DEET | − | − | + |
| diazinon | − | − | 0 |
| MCPA | − | − | + |
| pirimicarb | − | 0 | + |
| terbuthylazin | − | ± | − |
| diuron | − | − | ± |
| mecoprop | − | − | ± |
| terbutryn | − | − | ± |



**Table 5.** Signs of change for future physical parameters and micropollutant concentrations considering socio-economic scenarios only [assuming present climate+CurrPrac]. Notation is the same as for Tab. 4. +: general increase (red), −: general decrease (blue), ±: quantile-specific response (purple), 0: no change (grey)

| Variable | Moderate growth | Exploding growth | Decline |
|---|---|---|---|
| discharge | 0 | ± | 0 |
| water temperature | 0 | + | − |
| atrazin | 0 | − | 0 |
| isoproturon | 0 | − | 0 |
| metolachlor | 0 | − | 0 |
| 2,4-D | 0 | − | 0 |
| DEET | 0 | − | 0 |
| diazinon | 0 | − | 0 |
| MCPA | 0 | − | 0 |
| pirimicarb | 0 | − | 0 |
| terbuthylazin | 0 | − | 0 |
| diuron | + | + | 0 |
| mecoprop | + | + | 0 |
| terbutryn | + | + | 0 |

**Table 6.** Signs of change for future physical parameters and selected representative micropollutant concentrations considering management only [assuming present climate+Status quo]. The full version is shown in the SI (in section S5). Notation is the same as for Tab. 4. +: general increase (red), −: general decrease (blue), ±: quantile-specific response (purple), 0: no change (grey)

| Management measure | Q | T | Atrazin | Isoproturon | Diuron | Mecoprop |
|---|---|---|---|---|---|---|
| All | − | − | − | − | − | − |
| BanBioc | 0 | 0 | 0 | 0 | − | 0 |
| StoreVol | 0 | 0 | 0 | 0 | 0 | 0 |
| PermPave | − | 0 | 0 | 0 | + | + |
| RetRain | − | 0 | 0 | 0 | + | 0 |
| WWTP | 0 | 0 | 0 | 0 | − | − |
| OrgFarm | 0 | 0 | − | − | 0 | 0 |
| BufZone | 0 | − | − | − | 0 | 0 |
| NatPark | 0 | 0 | − | − | 0 | 0 |





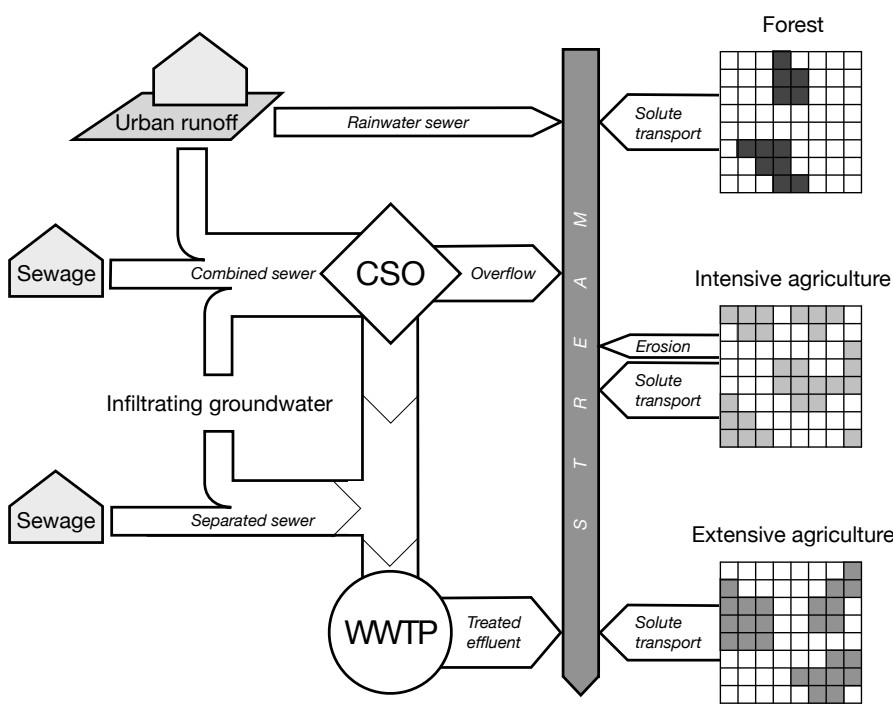

**Figure 1.** Pollutant sources, flow components and transport pathways considered in the iWaQa model.





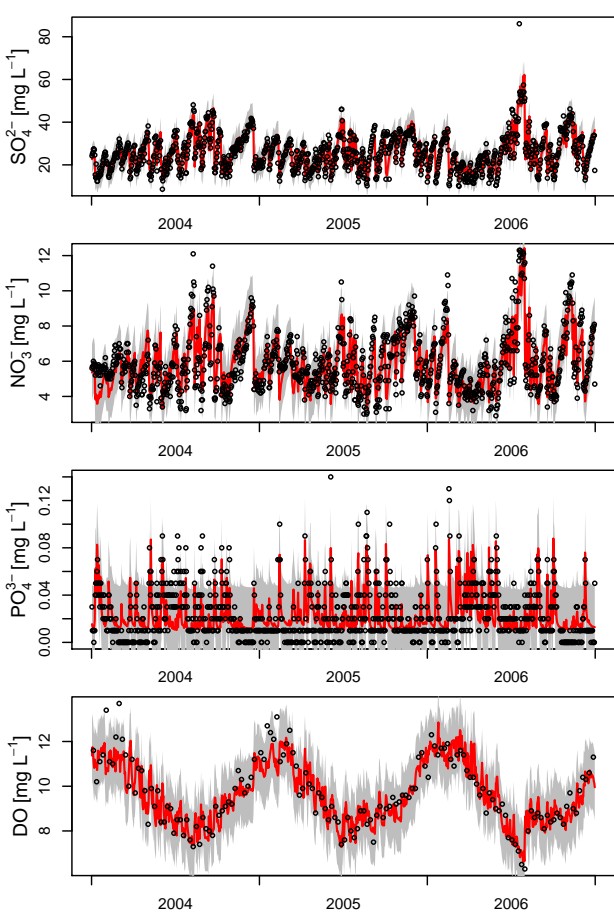

**Figure 2.** Calibration performance for selected inorganic parameters. Dots are observations, the red line is the model simulation with maximum posterior probability parameters, the grey region is the 95% uncertainty interval (total uncertainty). All values are daily mean concentrations.





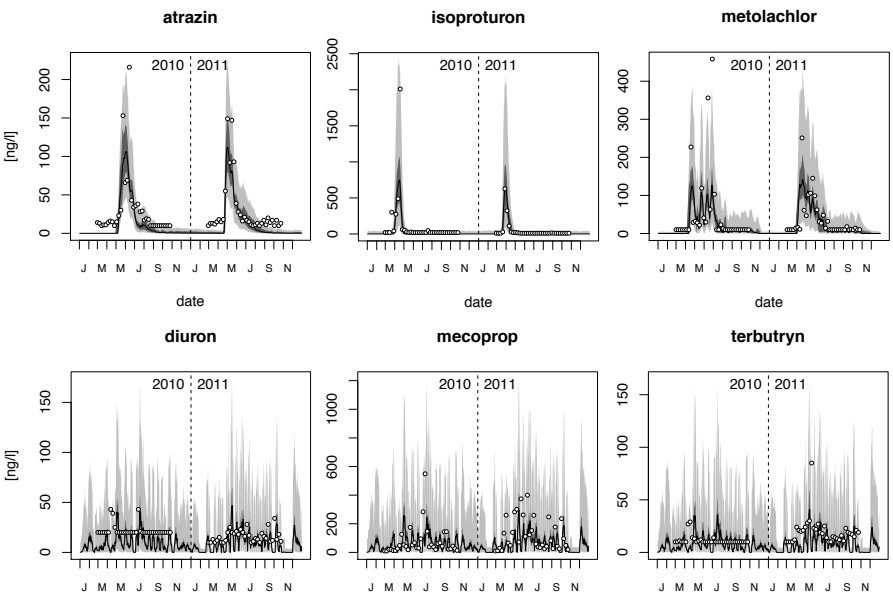

**Figure 3.** Calibration performance for selected micropollutants (top row: PPPs, bottom row: biocides). Dots are observations, black line is the model simulation with maximum posterior probability parameters, the grey region is the 95% uncertainty interval. All values are weekly mean concentrations; detection limits were 10 ng L-1 for all observations. PPPs not featured here are shown in section S3 in SI.

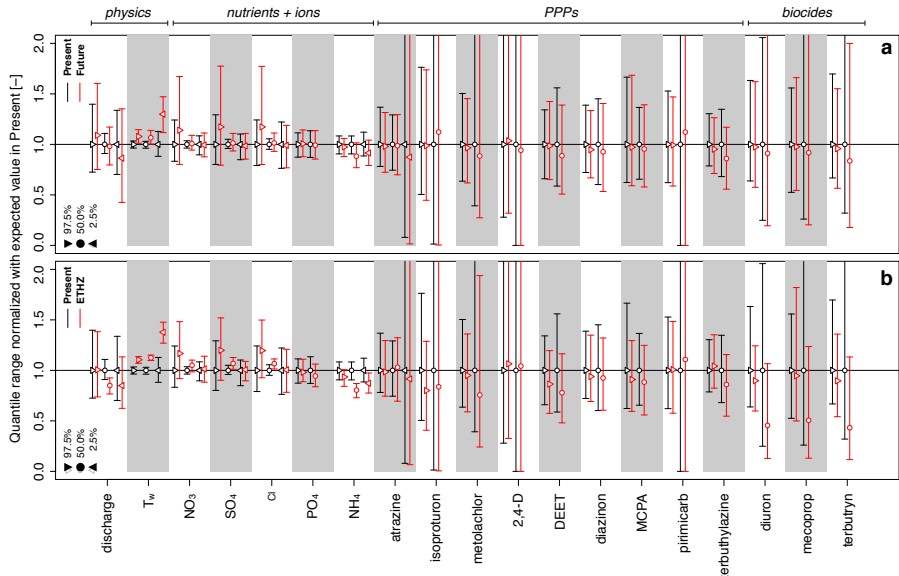

**Figure 4.** Relative change for high (Q97.5%), median (Q50%) and low (Q2.5%) quantiles of predicted model variables. a: Reference (calibration) period vs. ensemble future climate [assuming Status quo+CurrPrac in both cases]. b: Reference period vs. future according to the ETHZ HadCM3Q0 CLM model chain (drier climate) [assuming Status quo+CurrPrac in both cases]. Missing low quantiles correspond to cases where a present expected value of 0 for the low quantile prevented the meaningful normalisation of values.





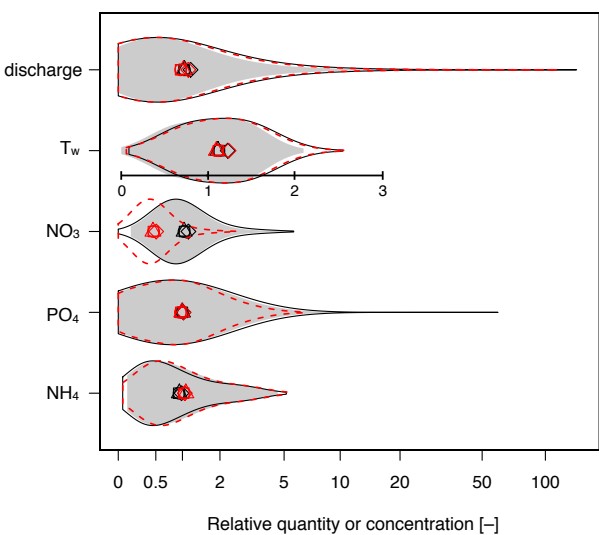

**Figure 5.** Ensemble future predictions for physical parameters and selected nutrients under management alternatives "All" (red) and "Cur-rPrac" (black), compared to present (grey). Distributions were normalized to be comparable across variables. Scale is logarithmic, except for water temperature, where the locally placed linear axis applies. Open symbols indicate mean concentrations of individual scenarios: circle: "Status quo", square: "Moderate growth", diamond: "Exploding growth", pyramid: "Decline".





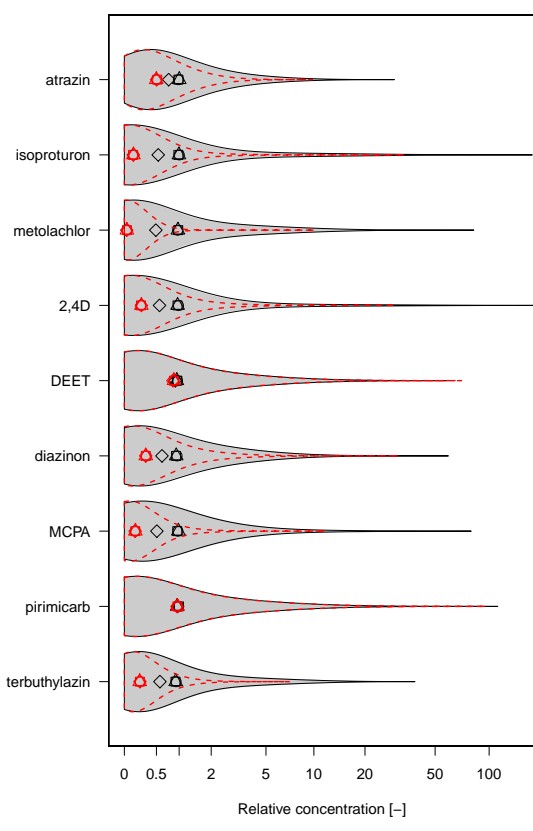

**Figure 6.** Ensemble future predictions for PPPs under management alternatives "All" (red) and "CurrPrac" (black), compared to present (grey). Distributions were normalized to be comparable across variables. Open symbols indicate mean concentrations of individual scenarios: circle: "Status quo", square: "Moderate growth", diamond: "Exploding growth", pyramid: "Decline".





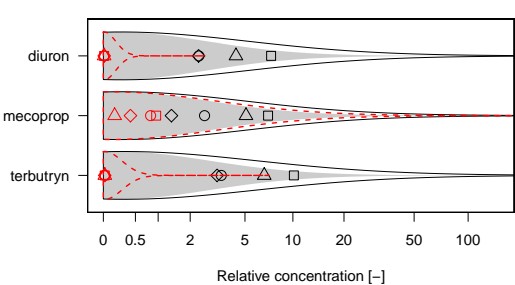

**Figure 7.** Ensemble future predictions for biocides under management alternatives "All" (red) and "CurrPrac" (black), compared to present (grey). Distributions were normalized to be comparable across variables. Open symbols indicate mean concentrations of individual scenarios: circle: "Status quo", square: "Moderate growth", diamond: "Exploding growth", pyramid: "Decline".