# Peer review of "Can integrative catchment management mitigate future water quality issues caused by climate change and socio-economic development?"

_Hydrology and Earth System Sciences, 2016_

## Referee Comment (RC1) · Anonymous Referee #1 · 26 Oct 2016

This paper describes a catchment water quality model that purports to satisfy the objectives of being both comprehensive and sufficiently simple for full uncertainty analysis. The model addresses both traditional water quality parameters, such as nutrients, as well as emerging contaminants, such as micropollutants. It also addresses features including climate change, population growth, migration, and socio-economic development. The model is demonstrated through application to a catchment in the Swiss plateau. Major results pertain to observations regarding the major sources of uncertainty as well as the relative importance of land use, management decisions, and climate change on water quality.

[Figure]

The manuscript is overall well-written, despite a few small grammatical and language errors, and is well-supported by figures and tables. The claim of a comprehensive model that can be subject to uncertainty analysis is certainly a potentially important contribution that is within the scope of HESS. However, it is unclear what novel breakthrough or insight the authors have made that allowed them to accomplish this contribution. In other words, what is it that allowed them to achieve this balance of completeness and simplicity when other previous researchers could not? The authors need to more clearly and convincingly explain this important aspect of their work. If they can do this, I believe the work is publishable in HESS.

Specific comments: 1. The abstract is unnecessarily long. I think it could be cut by a third to allow for a more concise overview. In particular, the first few sentences, which are mostly introductory content, could be removed without loss of information. Additionally, the abstract is rather vague near the end. The use of uncertainty quantification to inform robust management could be described more specifically.

2. The introduction is strong, with adequate citation of previous work and models. The last paragraph, however, should give an indication of HOW the main novelties of the paper were accomplished. What major task or insight allowed these contributions to be made?

3. My suspicion is that the separation of relevant flow components described in section 2.2 is one of the important contributions of this work, however it is described in too little detail for this to be clear.

4. How was the calibration accomplished? What calibration methods and criteria were used?

5. The results are described in good detail and are adequately supported by figures and tables. However, it would be valuable to describe how such results might actually be used. Robust management methods in response to high uncertainty is mentioned, but no specific examples are given. I think it is important to show how and why the

ability to perform uncertainty analysis adds value.

6. Section 5.1 includes the term "change signal" in the section title. What does this mean? This aspect is not actually described. At the top of page 14, it is described that "relative differences between different alternatives are often robust and can lead to a stable ranking of management alternatives." Is this what is meant by "change signal"? This point would be valuable to explore and to discuss. Might some of the large uncertainty in model predictions disappear if the focus is on differences between alternatives, rather than absolute predictions of alternatives?

7. In the conclusions, I found the first sentence of the second paragraph promising: "From our results we can derive definite recommendations for practical water management." However, I felt that the recommendations that followed (e.g. uncertainty analysis should always be performed, one should follow climate change effects) were not specific enough to be practical. I would have liked to have read more about how "proper accounting of uncertainty today" will "make management fairly robust" in the next decades. By what means is such robustness actually achieved? Overall, I feel the authors need to more strongly articulate their contribution in terms of the means by which they were able to accomplish their objectives. The paper would also be more valuable if they could demonstrate how the information on uncertainty would actually be used to formulate more robust management decisions.

---

## Referee Comment (RC2) · Anonymous Referee #2 · 5 Nov 2016

This paper describes the application of an integrated-water-resources-management model to a small catchment in the Swiss Plateau in order to analyze potential effects of different drivers on future water quality. The authors have chosen a question as title ("Can integrative catchment management mitigate future water quality issues caused by climate change and socio-economic development?"), but I am not sure whether they really answer it. It appears that some aspects of current practice (e.g., the application of pesticides within the catchment) are so uncertain that almost no scenario leads to significant changes, whereas other factors (in particular climate change in the study area) are simply not strong enough that a model could quantify effects. This is

somewhat frustrating – and I would actually have loved to get a clearer answer to the question posed in the title, which seems to be "no". Or maybe "not as long as we don't know what the farmers are actually applying on their fields".

I believe that the authors are stuck in a classical dilemma. They have chosen a study area with a comparably good data situation (even though the data in pesticide applications were still bad), in which not much direct climate change is expected. The temperature will rise, but won't make Northern Switzerland semi-arid. Precipitation estimates are highly uncertain and projections even don't agree on the sign of the change. Even before performing the study one could have guessed that climate change will have a smaller impact on water quality in the chosen catchment than changes in legislation regarding the use of certain pesticides. It would have been much more exciting to perform the analysis in a Mediterranean setting where much more severe climate-change effects can be expected, and maybe also more change in land use. Unfortunately, the data situation in these countries with respect to micropollutants is typically much worse than in Switzerland. Thus, while the authors' intension is a noble endeavor, their choice of application leads to some inconclusiveness.

I was not able to understand what the hydrological model does. Also the supplementary information is illusive in this regard. In particular it has been impossible for me to figure out how evapotranspiration is modeled, which in the study area most likely will more strongly be influenced by climate change than precipitation. The authors talk about three flow components (baseflow, subsurface flow, runoff), but it's not clear to me which physics are behind the separation between baseflow and subsurface flow. Neither could I see which flow component is how important. Maybe this has been described in one of the two preceding papers; but I have not read them.

The paper contains hardly any description of the catchment. However, the appropriateness of the model choice strongly depends on the geological setting and the agricultural land-use management. The authors need to explain this. My own quick research yielded: The geology is dominated by upper freshwater molasse (poorly per-

meable bedrock) with limited Quaternary overburden containing peat (particularly close to lake Greifensee). There is practically no groundwater body. The Mönchaltorfer Aa is connected to several drainage channels and/or tile drains, which exist essentially all over the valley. This information may justify a model concept that essentially denies explicit groundwater pathways. The infiltrating water is rapidly captured by the tile drains, so that restricting transformation processes to soil layers may be OK. However, other catchments are quite different and require an explicit treatment of groundwater flow, transport, and management. That is, integrated water resources management in the chosen catchment has the stream (and lake Greifensee) as its target, whereas in other catchments drinking-water production from groundwater is a major issue. In as much, "integrated water resources management" means different things in different catchments, requiring different model concepts.

The authors choose comparably simple descriptions for all processes, which I can understand given the difficulty of calibrating more complex models. However, if the transformation behavior of the pesticides is mainly based on calibrating simple first-order elimination models, the important influences of climate and land-use may be neglected. Some processes within the model contain an influence of air temperature (which is not identical with soil temperature). But I have not seen a potential influence of soil moisture, which may change more than precipitation in a warmer climate because of stronger evapotranspiration. Personally, I believe that including such effects will still not lead to strong climate signals in Northern-Swiss water quality. But, the best stochastic analysis does not help if the decisive dependencies are lacking in a model. Conversely, identifying the decisive dependencies requires data that cover a sufficient range in the controlling variables. In as much, I agree with some of the statements made by the authors in their discussions regarding the uncertainty caused by not identifying the decise mechanisms. While I don't have a solution either, it's not clear to me what the authors are recommending. Should we do more stochastic analysis on parameters that we can handle even though we know that the highest model uncertainty is on conceptual levels?

I am sorry that my (very much delayed) review contains remarks that are almost philosophical rather than going into specifics of the model and the application. But is appears that most of the latter was described in the two former papers. Therefore, it's also not easy to grasp what was actually done in this particular study without studying the preceding papers by the same authors.

---

## Editor Comment (EC1) · G. Characklis (Editor) · 10 Nov 2016

This paper has now received detailed and thoughtful comments from two reviewers. Both feel that the paper offers some promise, but that some significant issues will need to be addressed before it would be suitable for acceptance.

Reviewer #1's primary comments revolve mostly around the need to fully characterize the novel elements of the analysis. S/he asks for more clarity on exactly what is new in this analysis and how these advances enable greater insights.

Reviewer #2 has some questions regarding the ability of the analysis to answer the

question laid out in the title of the paper. S/he is concerned that the analysis described may not have the ability to isolate and identify some of the effects it seeks to characterize, and asks for a more detailed .

While both reviewers have a number of other comments and questions, the two above seem to describe their major concerns. These will not be trivial to address, and I hope that the authors will take the effort to address them in a complete and thoughtful manner. If they are able to do this, then it seems likely that this process will end with a positive final outcome.

---

## Author Comment (AC1) · 2 Dec 2016

We thank the reviewers for their constructive comments and the editor for highlighting the key issues. Below we respond to the comments one by one. Reviewers' comments are typeset in **bold**, responses in normal weight font.

[Figure]

**Review #1**

**The manuscript is overall well-written, despite a few small grammatical and language errors, and is well-supported by figures and tables. The claim of a comprehensive model that can be subject to uncertainty analysis is certainly a potentially important contribution that is within the scope of HESS. However, it is unclear what novel breakthrough or insight the authors have made that allowed them to accomplish this contribution. In other words, what is it that allowed them to achieve this balance of completeness and simplicity when other previous researchers could not? The authors need to more clearly and convincingly explain this important aspect of their work. If they can do this, I believe the work is publishable in HESS.**

Thank you for your appreciation. A grammar and language check will be carried out to remove errors.

In the revised version we will emphasize in the Introduction that the presented model framework was newly developed for this study and – except for the hydrological part – has not been published yet. This will be mentioned as point (1) of the novelties at the end of the Introduction. Based on the review of existing model frameworks we think that, first, integrating all relevant pollution pathways such as storm sewers and combined sewers (urban pollution), surface and underground diffuse pathways (agricultural pollution) into a parsimonious model and using it to predict micropollutant concentrations is novel. Second, the comprehensive uncertainty analysis provides insight regarding optimal decision support both on a conceptual level and for the case study area. This conceptual but parsimonious model system could simulate the observed large variety of pollutant dynamics by using only first-order controls of processes, which made it applicable to complex forecasts assuming changes in various boundary conditions. These novelties will be more explicitly mentioned in the Abstract, Discussion and Conclusions too.

**1. The abstract is unnecessarily long. I think it could be cut by a third to allow for a more concise overview. In particular, the first few sentences, which are mostly introductory content, could be removed without loss of information. Additionally, the abstract is rather vague near the end. The use of uncertainty quantification to inform robust management could be described more specifically.**

Thank you for this comment. If desired, we can streamline the introduction and further compress the rest of the abstract. The use of a quantification of uncertainty to inform robust management will be explained in more detail and the novelty of the model will be highlighted. We can thereby reduce the length from 383 to 350 words:

*"The design and evaluation of solutions for integrated surface water quality management requires an integrated modelling approach. Integrated models have to be comprehensive enough to cover the aspects relevant for management decisions, allow for mapping of larger-scale processes to the regional and local contexts. Besides this, models have to be sufficiently simple and fast to apply proper methods of uncertainty analysis, covering model structure deficits and error propagation through the chain of submodels. Here, we present a new integrated catchment model satisfying both conditions. The conceptual 'iWaQa' model was developed to support the integrated management of small streams. It can be used to predict traditional water quality parameters like nutrients and a wide set of organic micropollutants (plant and material protection products) by considering all major pollutant pathways in urban and agricultural environments. Due to its simplicity, the model allows for a full, propagative analysis of predictive uncertainty, including certain structural and input errors. The usefulness of the model is demonstrated by predicting future surface water quality in a small catchment with mixed land use in the Swiss Plateau. We consider climate change, population growth or decline, socio-economic development and the implementation of management strategies to tackle urban and agricultural point and non-point sources of pollution. Our results indicate that input and model structure uncertainties are the most influential factors for certain water quality parameters. In these cases*

[Figure]

*model uncertainty is high already for present conditions. Nevertheless, accounting for today's uncertainty makes management fairly robust to the foreseen range of potential changes in the next decades. The assessment of total predictive uncertainty allows selecting management strategies that show small sensitivity to poorly known boundary conditions. The identification of important sources of uncertainty helps to guide future monitoring efforts and pinpoints key indicators whose evolution should be closely followed to adapt management. The possible impact of climate change is clearly demonstrated by water quality substantially changing depending on single climate model chains. However, when all climate trajectories are combined the human land use and management decisions have a larger influence on water quality against a time-horizon of 2050 in the study."*

**2. The introduction is strong, with adequate citation of previous work and models. The last paragraph, however, should give an indication of HOW the main novelties of the paper were accomplished. What major task or insight allowed these contributions to be made?**

Reviewing the literature revealed that existing models are either too specific in scope or too complex in terms of structure to be building blocks of a comprehensive yet simple water quality model covering all major sources of pollution. Previous studies (e.g. Leu et al. 2004, Wittmer et al. 2010) mapped pathways of micropollutant transport into the river system and could be used to define the basic architecture of the present model. It was recognized, that the hydrological module published earlier (Honti et al. 2013, 2014) was not directly connectable to the water quality module due to differences in dominant processes regarding catchment hydrology and pollutant transport. To get an idea of pollutant hydrology, nutrients, oxygen, and selected ions were used to calibrate a flow partitioning scheme that divided total flow between pollutant transport pathways. This enabled us to get a good description of pollutant sources, including all three types of sewer systems (rain, combined, and separate), and the varying elimination efficiency at the wastewater treatment plant. Micropollutant data were not abundant, and without

sufficient temporal resolution to perform this step, but the different flow paths could be used in the micropollutant model afterwards. This approach allowed us to capture pollutant dynamics with very simple model structure, which was a crucial factor for making a proper uncertainty analysis.

The steps of this approach will be made clearer in the model description. Besides this, the introduction will be extended with an extra sentence before the objectives:

*"Hence, there is a need for water quality predictions, which are comprehensive enough to cover the aspects relevant for management decisions, allow for coupling of global change processes with the regional and local contexts, but are based on sufficiently simple models, which allow for proper uncertainty analysis that propagates uncertainty through the entire model chain. We recognised that these requirements can be fulfilled with a conceptual model framework that includes all major urban and agricultural pollution sources and transport pathways and Bayesian inference of catchment-specific parameters."*

**3. My suspicion is that the separation of relevant flow components described in section 2.2 is one of the important contributions of this work, however it is described in too little detail for this to be clear.**

We agree that this could be described in more detail. We will emphasize the novelty of calibration-based recognition of pollutant pathways and their binding to micropollutant transport and will provide a paragraph about this part of the model to section 2.2.

Basically, we sequentially applied a recursive baseflow filter (Eckhardt, 2005; Rimmer & Hartmann, 2014) to get slow (baseflow), fast (subsurface flow) and immediately (runoff on the rainfall's day) responding flow components. Separated flow components were distributed afterwards among pollution transport pathways based on the partitioning scheme calibrated based on the observed concentrations of traditional pollutants. Finally, these were coupled to micropollutant sources. We think that the key finding in this aspect is not the flow separation itself, it is rather the recognition that a hydrological model calibrated on total discharge will not be able to simulate the hydrologically negligible pathways delivering the majority of pollutants into streams and rivers.

**4. How was the calibration accomplished? What calibration methods and criteria were used?**

Section 2.2.3 will be extended with more details about the calibration method and objective functions. We used single-objective Bayesian calibration and uncertainty analysis based on formal statistical likelihood functions for all model outputs. Naturally, the likelihood functions differed by component, because a reliable likelihood function should reflect our knowledge about the error-generating processes, which are different from variable to variable. Hydrological predictions were mostly affected by input errors, therefore we could apply a likelihood function that can account for both input uncertainty and model structure deficits (del Giudice et al. 2012; Honti et al. 2013). Here a heteroscedastic, rainfall-dependent autoregressive function served as a Bayesian description for the impacts of these uncertainty sources in observed flow. The identification of error parameters was possible due to the huge number of good quality, frequent discharge data. Water quality data were more scarce, which prevented us from identifying the parameters of model bias and their input-dependence. Therefore we chose a simple i.i.d. normal error model for them. As residuals had strongly skewed distributions, we applied a Box-Cox transformation to better meet our assumption of normality.

**5. The results are described in good detail and are adequately supported by figures and tables. However, it would be valuable to describe how such results might actually be used. Robust management methods in response to high uncertainty is mentioned, but no specific examples are given. I think it is important to show how and why the ability to perform uncertainty analysis adds value.**

To demonstrate the relevance for management more explicitly, we will add a paragraph about how the results can be used in a multi-criteria decision support process and why

it is important to quantify the uncertainty of predictions:

*"The model results can directly be used by river managers for a cost-benefit (Hanley & Spash, 1993) or a multi-criteria decision support analysis (Reichert et al. 2015). Which approach is most appropriate depends on the management objective. For example, if the goal is to achieve a good chemical state of the river by a specific point in time, predictions can be fed into the chemical assessment procedure and the managers can screen for the management alternatives with the highest probability to achieve a good state considering all future scenarios. If the management objective is to choose the most effective management alternatives given a fixed budget, the managers can screen first for all (combinations of) measures that meet the budget and then select the most effective ones. In general, providing predictive uncertainty in addition to the best guess of consequences of management alternatives is crucial (1) to assess if differences between alternatives are significant, (2) for the search of alternatives that are robust regarding uncertain changes in future boundary conditions and (3) to support credibility of scientific research by being transparent about predictive uncertainty. In contrast to intuition, a high forecast uncertainty does not preclude effective decision support, because it often does not affect the relative ranking of competing management alternatives (Reichert & Borsuk, 2005). Last, but not least, quantified uncertainty (4) facilitates learning by updating the predictions when new information arises (e.g. by monitoring changes after implementation of measures)."*

In addition, we make it explicit why elucidating the sources of predictive uncertainty is relevant for stakeholders (see response to comment 7 below).

**6. Section 5.1 includes the term "change signal" in the section title. What does this mean? This aspect is not actually described. At the top of page 14, it is described that "relative differences between different alternatives are often robust and can lead to a stable ranking of management alternatives." Is this what is meant by "change signal"? This point would be valuable to explore and to discuss. Might some of the large uncertainty in model predictions disappear if the**

**focus is on differences between alternatives, rather than absolute predictions of alternatives?**

Thank you for spotting this mismatch. We simply used the wrong term. We will replace 'change signal' by 'future change', because that is the focus of that section. Regarding a stable ranking of management alternatives, we could not set up a ranking here: this being an exposure study, we had no information on the costs of alternatives and preferred endpoints of the involved stakeholders (this will be explored in a following stage). The sentence in the text *"... although absolute water quality changes might be very uncertain, relative differences between different alternatives are often robust and often lead to a stable ranking of management alternatives"* just wanted to highlight that high uncertainty doesn't ab ovo preclude decision making.

**7. In the conclusions, I found the first sentence of the second paragraph promising: "From our results we can derive definite recommendations for practical water management." However, I felt that the recommendations that followed (e.g. uncertainty analysis should always be performed, one should follow climate change effects) were not specific enough to be practical. I would have liked to have read more about how "proper accounting of uncertainty today" will "make management fairly robust" in the next decades. By what means is such robustness actually achieved? Overall, I feel the authors need to more strongly articulate their contribution in terms of the means by which they were able to accomplish their objectives. The paper would also be more valuable if they could demonstrate how the information on uncertainty would actually be used to formulate more robust management decisions.**

We agree and we will discuss how management decisions may differ based on the sources of uncertainty:

When only total uncertainty is known (without knowing anything about its sources) there is no information on how to reduce uncertainty. In such a case, a management

action can be considered robust if it shows weak sensitivity to poorly known boundary conditions (e.g. it always improves certain parameters). However, when total uncertainty is attributed to certain input data, boundary conditions, or sub-models, an iterative adaptive management process focusing on the critical aspects can be designed:

- Input uncertainty: if reducing input uncertainty would be worth the efforts for getting better input data, managers should invest in getting such data. Hence, there is an action identified that can be actively taken.

- Model uncertainty: the same as before. If process understanding etc. limits the decision making, effort (research) need to go into a better representation by better models and data.

- Climate change uncertainty: if the scenario analyses reveal that future climate change uncertainty is the most limiting predictions there is no use of improving input data or akin. However, it might be recommended to carefully monitor how climate actually develops and how predictions perform over time such that the evaluations of management options may improve over time. In this case, a re-assessment after a couple of years might be recommended.

Overall, depending on the outcome of the uncertainty analysis, different recommendations can be given to stakeholders. In our case the practical recommendations are summarised in the reply to comment #1 of Reviewer #2. Besides the ranking of boundary conditions, we identified that in our case study input and model-related uncertainties dominated for most micropollutants, so monitoring and research efforts should be concentrated on them unless a clear political willing arises to ban certain micropollutant groups on the catchment. This will be made explicit in the conclusions.

**Review #2**

**This paper describes the application of an integrated-water-resources-management model to a small catchment in the Swiss Plateau in order to analyze potential effects of different drivers on future water quality. The authors have chosen a question as title ("Can integrative catchment management mitigate future water quality issues caused by climate change and socio-economic development?"), but I am not sure whether they really answer it. It appears that some aspects of current practice (e.g., the application of pesticides within the catchment) are so uncertain that almost no scenario leads to significant changes, whereas other factors (in particular climate change in the study area) are simply not strong enough that a model could quantify effects. This is somewhat frustrating – and I would actually have loved to get a clearer answer to the question posed in the title, which seems to be "no". Or maybe "not as long as we don't know what the farmers are actually applying on their fields".**

We agree that this needs a clarification and a more explicit answer to the question in the title. The answer will be: *"Despite all uncertainties, the results clearly show that the direct human influence via the local, regional and national scales exerts the largest effect on water quality, so the answer to the question whether management could mitigate future water quality issues was generally positive for our case study."*

We will revise the paper to provide a clear explanation for our conclusion.

Most micropollutant management scenarios caused an observable change, but change was always associated with high uncertainty. The same applies to climate at the level of individual model chains, but not climate change as a whole. Based on our experience, it seems quite rare in environmental modeling that one gets a significant change in future predictions in terms of a large shift to a different state with small variability. Moreover, statistical significance has multiple facets and definitions, a small shift in mean can be significant despite any high variability when enough past (observed) and

future (predicted) points support it. It is true that we can model micropollutants only with high uncertainty, but we are quite certain in what direction management would change their environmental concentration, and therefore the preferred way of management is much more certain than the prediction of the next concentration peak.

The answer to the question for our case study is generally 'YES', but with some amendments: Socio-economic scenarios in this study were often really influential compared to climate change, but they were compiled by local stakeholders on the ground of reality, just like the climate predictions by the creators of the ENSEMBLES database. Moreover, climate change as a whole is more uncertain than the individual model chains due to the divergence between them. But the same applies to future socio-economic development: if we pour all 4 scenarios together (symbolizing that we don't know which will actually happen), clear individual changes blur into uncertainty. The overall answer is that most management measures were proven to be powerful enough to compensate for non-manageable effects, such as climate change and socio-economic development. Therefore, with careful planning and continuous monitoring of the direction of socio-economic development, one can actually maintain the present water quality and even improve certain components.

**I believe that the authors are stuck in a classical dilemma. They have chosen a study area with a comparably good data situation (even though the data in pesticide applications were still bad), in which not much direct climate change is expected. The temperature will rise, but won't make Northern Switzerland semi-arid. Precipitation estimates are highly uncertain and projections even don't agree on the sign of the change. Even before performing the study one could have guessed that climate change will have a smaller impact on water quality in the chosen catchment than changes in legislation regarding the use of certain pesticides. It would have been much more exciting to perform the analysis in a Mediterranean setting where much more severe climate-change effects can be expected, and maybe also more change in land use. Unfortunately, the data situ-**

**ation in these countries with respect to micropollutants is typically much worse than in Switzerland. Thus, while the authors' intension is a noble endeavor, their choice of application leads to some inconclusiveness.**

First, we would like to mention that we did not aim at selecting the study area such that getting a very clear, "conclusive" outcome. Neither were the results evident from the outset. The project was part of the Swiss National Research Program on Sustainable Water Management (NRP 61; www.nrp61.ch) that asked how different driving factors such as climate change or socio-economic development may affect water resources across Switzerland. It follows from that that the study region is not an exceptional case; in the contrary it represents a typical situation where the respective influence of different factors is not very obvious. We think that answers to such an important question are of an undeniable scientific interest. According to our results the study area will clearly suffer from a substantial climate change effect. Individual model chains predict a significant warming and shifts in the current precipitation regime. What blurs this effect is first the discrepancy between individual model chains that converts strong but contradictory signals into uncertainty (especially because of the precipitation forecasts) and second the dynamic socio-economic setting that renders dramatic changes in population and land-use feasible. Individual model chains show clear increase or decrease in the amount of annual precipitation by up to 30%, which is clearly significant. The situation of model chains not agreeing even on the sign of precipitation change is however not special, it applies to most of Central Europe, where Mediterranean, Atlantic and even Continental climatic influence varies within a year. The climatic modeling of this huge climatic transition zone is even more uncertain than elsewhere (Switzerland may be even more complicated due to the influence of the Alps). So while our chosen study site is not optimal in terms of the obviousness of precipitation trends, it represents a large and important region, including the densely populated Swiss Plateau.

We will rephrase the results section on climate change (4.3) to emphasize that climate change is strong but uncertain: *"However, because only one climate change path will*

[Figure]

*develop, the analysis of single model chains proves that water quality may change substantially depending on the actual future climate."*

Legislative management measures are certainly powerful, but they are not omnipotent. A large portion of existing uncertainty of pesticide modeling originates from violations from the existing rules. Illegal applications for unknown purpose and spills probably due to improper handling produce huge concentration peaks for many compounds that are impossible to explain on grounds of regulations and good practice. This uncertainty was actually propagated through our models, and therefore it often happened that predicted concentrations were not 0 even after a future ban. In this aspect our modeling added some details to the obvious: a ban is effective, but will not eliminate all problems.

**I was not able to understand what the hydrological model does. Also the supplementary information is illusive in this regard. In particular, it has been impossible for me to figure out how evapotranspiration is modeled, which in the study area most likely will more strongly be influenced by climate change than precipitation. The authors talk about three flow components (baseflow, subsurface flow, runoff), but it's not clear to me which physics are behind the separation between baseflow and subsurface flow. Neither could I see which flow component is how important. Maybe this has been described in one of the two preceding papers; but I have not read them.**

Indeed, we did not include a full description of the hydrological model in this publication, because this simple (and popular) model has been described in at least 3 other papers (Kuczera et al. 2006, Honti et al. 2013, Honti et al. 2014). We surely could add it to the supporting information, but would only do so at the explicit request of the editor. Details about the flow separation will be added to the text (see answer to Reviewer #1).

For the record, evapotranspiration was modelled from potential evapotranspiration (calculated by the Hargreaves-Samani method [Hargreaves & Samani, 1982]) and the actual moisture content of the topsoil. Evapotranspiration will not dramatically change in

the future as the region is humid and actual and potential ET are quite close to each other. As the reviewer suggests between the lines, this is clearly different in warmer and arid regions.

The separation of flow components is rather conceptual, just like the hydrological model itself. Runoff, subsurface flow and baseflow indicate instant (e.g. on the same day as the precipitation), fast (few days after precipitation), and slowly reacting flow components. No physical validation of this separation was possible due to lack of data. Flow separation had to be introduced because we had to use the actual measured flow for the calibration of flow distribution among pollution pathways (based on observed concentrations of traditional pollutants), as the otherwise well-performing hydrological model still had a non-negligible uncertainty, which could have spoiled the calibration. Afterwards in the prediction phase, to maintain compatibility with the calibration phase, we used the same separation algorithm on the simulated total flow.

**The paper contains hardly any description of the catchment. However, the appropriateness of the model choice strongly depends on the geological setting and the agricultural land-use management. The authors need to explain this. My own quick research yielded: The geology is dominated by upper freshwater molasse (poorly permeable bedrock) with limited Quaternary overburden containing peat (particularly close to lake Greifensee). There is practically no groundwater body. The Mönchaltorfer Aa is connected to several drainage channels and/or tile drains, which exist essentially all over the valley. This information may justify a model concept that essentially denies explicit groundwater pathways. The infiltrating water is rapidly captured by the tile drains, so that restricting transformation processes to soil layers may be OK. However, other catchments are quite different and require an explicit treatment of groundwater flow, transport, and management. That is, integrated water resources management in the chosen catchment has the stream (and lake Greifensee) as its target, whereas in other catchments drinking-water production from groundwater is a major issue.**

**In as much, "integrated water resources management" means different things in different catchments, requiring different model concepts.**

We agree that better describing the catchment improves the clarity of the presentation. We will also extend the catchment description to clarify the role of groundwater flow in the study catchment. Actually there is an influential groundwater body in the catchment supplying the baseflow in the local streams. Tile drains cover only a part of the catchment and are quite shallow. The model includes groundwater contributions to flow and transport pathways via them, but for most micropollutants we considered the travel time through the unsaturated and the saturated zone so long that the compounds had undergone substantial degradation causing low concentration in groundwater. This assumption was supported by measurements demonstrating low background concentrations of these chemicals during baseflow conditions.

While we think that the approach is quite general, we agree that transferring this conceptual model into another catchment would certainly require a recalibration and possibly also a structural revision (if system analysis reveals that important components are missing). This we mention under section 5.3. However, the model was intended to simulate surface water quality, and therefore it's certainly not capable of predicting groundwater (for drinking water production) or soil quality. We will make this clearer already in the abstract.

**The authors choose comparably simple descriptions for all processes, which I can understand given the difficulty of calibrating more complex models. However, if the transformation behavior of the pesticides is mainly based on calibrating simple first- order elimination models, the important influences of climate and land-use may be neglected. Some processes within the model contain an influence of air temperature (which is not identical with soil temperature). But I have not seen a potential influence of soil moisture, which may change more than precipitation in a warmer climate be- cause of stronger evapotranspiration. Personally, I believe that including such effects will still not lead to strong climate**

**signals in Northern-Swiss water quality. But, the best stochastic analysis does not help if the decisive dependencies are lacking in a model. Conversely, identifying the decisive dependencies requires data that cover a sufficient range in the controlling variables. In as much, I agree with some of the statements made by the authors in their discussions regarding the uncertainty caused by not identifying the decise mechanisms. While I don't have a solution either, it's not clear to me what the authors are recommending. Should we do more stochastic analysis on parameters that we can handle even though we know that the highest model uncertainty is on conceptual levels?**

Our pesticide model is strongly conceptual, and therefore it relies heavily on calibration. The otherwise weakly known intricacies of pesticides' biotransformation in soils are encapsulated in the first-order decay rates. As the region is expected to remain humid in the future, we don't anticipate huge changes in these empirical rates. The model is certainly not applicable to other regions without recalibration and possibly a structural review. In a region where soil moisture actually limits biotransformation the calibration database would contain data revealing this phenomenon and therefore it would either be assimilated into the decay rates or make obvious that such dependency needs to be implemented in the model. These issues are discussed in section 5.3.

It's right that the stochastic analysis can't mend the model. However, a proper uncertainty analysis will translate model deficiencies into stochastic variability and reveal the true uncertainty of model predictions to the decision maker. A good practice seems to be periodically analyzing key system properties that indicate or even foretell system transitions. This will be integrated into the conclusions as parts of the practical recommendations and the outline on how uncertainty analysis can improve management.

**I am sorry that my (very much delayed) review contains remarks that are almost philosophical rather than going into specifics of the model and the application. But is appears that most of the latter was described in the two former papers. Therefore, it's also not easy to grasp what was actually done in this particular**

**study without studying the preceding papers by the same authors.**

The entire water quality model presented here is novel, we will emphasize this novelty more (please see responses to Reviewer #1). Depending on the Editor's opinion about the length of the manuscript, we are happy to include more details about the models into the main text.

**References**

del Giudice, D., M. Honti, A. Scheidegger, C. Albert, P. Reichert, és J. Rieckermann (2013), Improving uncertainty estimation in urban hydrological modeling by statistically describing bias, *Hydrol. Earth Syst. Sci.*, *17*(10), 4209–4225, doi: 10.5194/hess-17-4209-2013.

Eckhardt, K.: How to construct recursive digital filters for baseflow separation, Hydrol. Process., 19, 507–515, doi: 10.1002/hyp.5675, 2005.

Hanley, N., and C. L. Spash: Cost-benefit analysis and the environment. Aldershot, Hants, England: E. Elgar. 1993.

Hargreaves, G. and Z. Samani: Reference crop evapotranspiration from temperature, Transactions of ASAE, 1, 96 – 99, 1982.

Honti, M., C. Stamm, and P. Recihert: Integrated uncertainty assessment of discharge predictions with a statistical error model, Water Resour. Res., 49, 4866 – 4884, doi: 10.1002/wrcr.20374, url: http://dx.doi.org/10.1029/2009WR008884, 2013.

Honti, M., A. Scheidegger, and C. Stamm: The importance of hydrological uncertainty assessment methods in climate change impact studies, Hydrol. Earth Syst. Sci., 18, 3301–3317, doi: 10.5194/hess-18-3301-2014, 2014.

Kuczera, G., D. Kavetski, S. Franks, and M. Thyer (2006), Towards a Bayesian total

error analysis of conceptual rainfall-runoff models: Characterising model error using storm-dependent parameters, *Journal of Hydrology*, *331*(1-2), 161 – 177.

Leu, C., H. Singer, C. Stamm, S. R. Müller, and R. P. Schwarzenbach: Simultaneous Assessment of Sources, Processes, and Factors Influencing Herbicide Losses to Surface Waters in a Small Agricultural Catchment, Environmental Science & Technology, 38, 3827–3834, doi: 10.1021/es0499602, 2004.

Reichert, P. and M. Borsuk: Does high forecast uncertainty preclude effective decision support?, Environmental Modelling & Software, 20, 991–1001, doi: 10.1016/j.envsoft.2004.10.005, 2005.

Reichert, P., S. D. Langhans, J. Lienert, and N. Schuwirth: The conceptual foundation of environmental decision support, Journal of Environmental Management, 154, 316–332, doi: 10.1016/j.jenvman.2015.01.053, 2015.

Rimmer, A. and A. Hartmann: Optimal hydrograph separation filter to evaluate transport routines of hydrological models, Journal of Hydrology, 514, 249–257, doi: 10.1016/j.jhydrol.2014.04.033, 2014.

Wittmer, I., Bader, H.-P., Scheidegger, R., Singer, H., Lück, A., Hanke, I., Carlsson, C., and Stamm, C.: Significance of urban and agricultural land use for biocide and pesticide dynamics in surface waters, Water Research, 44, 2850 – 2862, doi: 10.1016/j.watres.2010.01.030, 2010.

---

## Author Response (AR1)

We thank the reviewers and the editor for their constructive comments. In the revised version we carried out all modifications proposed in the initial response to the reviews and addressed the two comments of the editor as well. In the following we introduce the changes we made following each comment. Comments are typeset in **bold**, responses in normal weight font.

**Editor's comments**

5    **1) I believe that some description of the hydrologic model that serves as the basis for this work, and is published in earlier work, would be a useful addition to this paper (as requested by Reviewer #2). This can be a general description which does not need to be more than 1-2 paragraphs, and can be placed in the supplemental information (although it might fit better in the main body of the paper). I believe this will be helpful to readers, who can then access the earlier papers if they would like more detail.**

10    The description of the hydrologic model was revised and expanded to be more informative on the basics of this simple conceptual model:

*"We use the modified LogSPM conceptual model (original: Kuczera et al. [2006], modified version: Honti et al. [2013, 2014]) to simulate stream discharge at the catchment outlet due to its simplicity and acceptable performance of its varieties in several catchments (Kuczera et al. 2006, Reichert and Mieleitner 2009, Honti et al. 2013). This conceptual model belongs to the*
15    *saturated path family of models, it describes runoff formation by assuming a non-linear function that unambiguously maps between average soil moisture content and saturated fraction of catchment surface (Kavetski et al. 2003). We use the modified parameterisation by Honti et al. (2013) for the saturation function, which is based on the catchment-scale analogies of characteristic soil moisture contents. Field capacity corresponds to soil moisture when saturated area is negligible, while full saturation means an almost complete saturation of the catchment area. Water is routed from the soil storage to either surface*
20    *runoff, subsurface flow or recharge, proportionally to the saturated area. A linear groundwater storage produces the long-term memory of the model and provides baseflow. Surface runoff, subsurface flow and baseflow make up the total flow in the stream. A detailed description of the model and calibration procedure can be found in Honti et al. (2013, 2014)."*

**(2) a more detailed description of the geology and hydrology in the study region would be a useful addition, and also provide more credibility in terms of the application of the proposed methods to the circumstances involved. Some mention**
25    **of the both the hydrogeological details and the human modifications (e.g., tile drains) would help readers understand why the proposed methods are suitable in this case, as well as providing some indication as to how generalizable these methods might be to other regions.**

A short description of the geology and hydrology of the site was added:

*"Geologically, the area is dominated by tertiary river deposits (Upper Freshwater Molasse) and moraines from the last glacia-*
30    *tion (Würm) (Zingg, 1934; https://map.geo.admin.ch). Both have a rather poor permeability. From these substrates, cambisols and gleysols have developed as the main soil types on the hillslopes and flat areas, respectively (FAL, 1996). About 50% of agricultural soils are artificially drained by tile and pipe drains. The area is characterized by shallow groundwater that feeds the baseflow throughout stream network."*

An extra sentence was added to section 5.3 about the need to adapt the model parameters and even the structure when applying
35    it in other catchments:

*"Therefore, while the approach is quite general, transferring this conceptual model into another catchment would certainly require a recalibration. If system analysis reveals that important components are missing, the model structure needs to be adapted, too."*

**Review #1**

**The manuscript is overall well-written, despite a few small grammatical and language errors, and is well-supported by figures and tables. The claim of a comprehensive model that can be subject to uncertainty analysis is certainly a potentially important contribution that is within the scope of HESS. However, it is unclear what novel breakthrough or insight the authors have made that allowed them to accomplish this contribution. In other words, what is it that allowed them to achieve this balance of completeness and simplicity when other previous researchers could not? The authors need to more clearly and convincingly explain this important aspect of their work. If they can do this, I believe the work is publishable in HESS.**

A grammar and language check was carried out and several grammatical and spelling errors were removed.

In the revised version we emphasize in the Introduction that the presented model framework was newly developed for this study and – except for the hydrological part – has not been published yet. The introduction now makes clear the motivations of developing the new model:

*"Hence, there is a need for water quality predictions, which are comprehensive enough to cover the aspects relevant for management decisions, allow for coupling of global change processes with the regional and local contexts, but are based on sufficiently simple models, which allow for proper uncertainty analysis that propagates uncertainty through the entire model chain. We recognised that these requirements can be fulfilled with a conceptual model framework that includes all major urban and agricultural pollution sources and transport pathways and Bayesian inference of catchment-specific parameters."*

The following sentence emphasizes the novelty of the model:

*"First, we introduce a new conceptual catchment model (the iWaQa model) that was developed for small streams with IWRM-specific objectives in mind (simple, consistent and comprehensive in terms of both pollutants and pollution sources). Due to the model's simplicity, it is possible to estimate parameters from data and to perform a nonlinear error propagation with Monte Carlo techniques in a 'total uncertainty analysis' framework, which overcomes the major limitations mentioned above."*

In the abstract it was made clear that covering all major pollutant sources while being simple enough to be subject of a full uncertainty analysis are the key aspects of the new model:

*"The conceptual 'iWaQa' model was developed to support the integrated management of small streams. It can be used to predict traditional water quality parameters like nutrients and a wide set of organic micropollutants (plant and material protection products) by considering all major pollutant pathways in urban and agricultural environments. Due to its simplicity, the model allows for a full, propagative analysis of predictive uncertainty, including certain structural and input errors."*

**1. The abstract is unnecessarily long. I think it could be cut by a third to allow for a more concise overview. In particular, the first few sentences, which are mostly introductory content, could be removed without loss of information. Additionally, the abstract is rather vague near the end. The use of uncertainty quantification to inform robust management could be described more specifically.**

The abstract's introduction part was streamlined and an explicit description of using the results for robust management was added. The abstract's overall length was reduced from 383 to 354 words:

*"The design and evaluation of solutions for integrated surface water quality management requires an integrated modelling approach. Integrated models have to be comprehensive enough to cover the aspects relevant for management decisions, allow for mapping of larger-scale processes such as climate change to the regional and local contexts. Besides this, models have to be sufficiently simple and fast to apply proper methods of uncertainty analysis, covering model structure deficits and error propagation through the chain of submodels. Here, we present a new integrated catchment model satisfying both conditions. The conceptual 'iWaQa' model was developed to support the integrated management of small streams. It can be used to predict traditional water quality parameters like nutrients and a wide set of organic micropollutants (plant and material protection products) by considering all major pollutant pathways in urban and agricultural environments. Due to its simplicity, the model*

*allows for a full, propagative analysis of predictive uncertainty, including certain structural and input errors. The usefulness of the model is demonstrated by predicting future surface water quality in a small catchment with mixed land use in the Swiss Plateau. We consider climate change, population growth or decline, socio-economic development and the implementation of management strategies to tackle urban and agricultural point and non-point sources of pollution. Our results indicate that input and model structure uncertainties are the most influential factors for certain water quality parameters. In these cases model uncertainty is high already for present conditions. Nevertheless, accounting for today's uncertainty makes management fairly robust to the foreseen range of potential changes in the next decades. The assessment of total predictive uncertainty allows selecting management strategies that show small sensitivity to poorly known boundary conditions. The identification of important sources of uncertainty helps to guide future monitoring efforts and pinpoints key indicators whose evolution should be closely followed to adapt management. The possible impact of climate change is clearly demonstrated by water quality substantially changing depending on single climate model chains. However, when all climate trajectories are combined the human land use and management decisions have a larger influence on water quality against a time-horizon of 2050 in the study."*

**2. The introduction is strong, with adequate citation of previous work and models. The last paragraph, however, should give an indication of HOW the main novelties of the paper were accomplished. What major task or insight allowed these contributions to be made?**

The following sentences were added to the introduction to highlight our motivations for developing the new model:

*"Hence, there is a need for water quality predictions, which are comprehensive enough to cover the aspects relevant for management decisions, allow for coupling of global change processes with the regional and local contexts, but are based on sufficiently simple models, which allow for proper uncertainty analysis that propagates uncertainty through the entire model chain. We recognised that these requirements can be fulfilled with a conceptual model framework that includes all major urban and agricultural pollution sources and transport pathways and Bayesian inference of catchment-specific parameters."*

The model description part was amended at several places to better introduce the train of thought that led to the presented model structure.

**3. My suspicion is that the separation of relevant flow components described in section 2.2 is one of the important contributions of this work, however it is described in too little detail for this to be clear.**

We emphasize the novelty of calibration-based recognition of pollutant pathways and their binding to micropollutant transport and will provide a paragraph about this part of the model to section 2.2.

The first paragraph of section 2.2 already introduced the motivation to use a more detailed flow separation than the one in the hydrological model:

"The hydrological needs of a conceptual model aiming to simulate pollutant transport are usually different from the needs of a hydrological catchment model. Due to the specific transport pathways of pollutants one needs to consider water fluxes that are extremely important for the propagation of a given pollutant family but may not be of any particular importance for streamflow on the catchment or sub-catchment scale. This means that – building on the modelled flow components – we have to use a more elaborate flow routing scheme that includes all important pollutant transport pathways but remains easily derivable from the catchment-scale flow components (baseflow, subsurface flow, runoff)."

In the same section more details were added to the description of the flow separation routine:

*"We divide catchment-scale flow components between more detailed flow paths according to simple linear partitioning rules (section S2 in the SI). First we sequentially applied a recursive baseflow filter (Eckhardt, 2005; Rimmer & Hartmann, 2014) to get slow (baseflow), fast (subsurface flow) and immediately (runoff on the rainfall's day) responding flow components. Separated flow components were distributed afterwards among pollution transport pathways based on the partitioning scheme calibrated based on the observed concentrations of nutrients and major ions (see section 3.3)."*

Whereas in section 3.3 we describe how the further separated flow components (after calibrating to observed ion concentrations) could be used to model the pathways of micropollutants:

*"This flow partitioning enabled us to get a good description of the various pollutant sources, including all three types of sewer systems (rain, combined, and separate), and the varying elimination efficiency at the wastewater treatment plant. Micropollu-*
5 *tant data were less abundant, and without sufficient temporal resolution to be involved in the recognition of pollutant pathways, but the established flow paths could be used in the micropollutant model afterwards."*

**4. How was the calibration accomplished? What calibration methods and criteria were used?**

The first paragraph of section 2.2.3 was rewritten to provide more information on the calibration process:

*"The uncertainty of model and error parameters was assessed with single-objective Bayesian calibration and uncertainty anal-*
10 *ysis based on formal statistical likelihood functions for all model outputs. Sub-models were calibrated separately. Naturally, the likelihood functions differed by component, because a reliable likelihood function should reflect our knowledge about the error-generating processes, which are different from variable to variable. Hydrological predictions were mostly affected by input errors, therefore we could apply a likelihood function that can account for both input uncertainty and model structure deficits (del Giudice et al. 2013, Honti et al. 2013). Here a heteroscedastic, rainfall-dependent autoregressive function served*
15 *as a Bayesian description for the impacts of these uncertainty sources in observed flow. The identification of error parameters was possible due to the huge number of good quality, frequent discharge data. For water temperature data and traditional pol-lutants a simpler error model (independent, identically distributed normal errors) was applied satisfactorily, as data scarcity prevented us from identifying the parameters of model bias and their input-dependence. The same error model was used for mi-cropollutants, but a Box-Cox transformation with an exponent of 0.3 was applied to both the observed and modelled time-series*
20 *to account for heteroscedasticity originating from the high variability of these data. "*

**5. The results are described in good detail and are adequately supported by figures and tables. However, it would be valuable to describe how such results might actually be used. Robust management methods in response to high uncertainty is mentioned, but no specific examples are given. I think it is important to show how and why the ability to perform uncertainty analysis adds value.**

25 To demonstrate the relevance for management more explicitly, we added a paragraph to section 4.6 about the potential use of results in a multi-criteria decision support process and the importance of quantifying the uncertainty of predictions:

*"The model results can directly be used by river managers for a cost-benefit (Hanley & Spash, 1993) or a multi-criteria deci-sion support analysis (Reichert et al. 2015). Which approach is most appropriate depends on the management objective. For example, if the goal is to achieve a good chemical state of the river by a specific point in time, predictions can be fed into the*
30 *chemical assessment procedure and the managers can screen for the management alternatives with the highest probability to achieve a good state considering all future scenarios. If the management objective is to choose the most effective management alternatives given a fixed budget, the managers can screen first for all (combinations of) measures that meet the budget and then select the most effective ones. In general, providing predictive uncertainty in addition to the best guess of consequences of management alternatives is crucial (1) to assess if differences between alternatives are significant, (2) for the search of*
35 *alternatives that are robust regarding uncertain changes in future boundary conditions and (3) to support credibility of scien-tific research by being transparent about predictive uncertainty. In contrast to intuition, a high forecast uncertainty does not preclude effective decision support, because it often does not affect the relative ranking of competing management alternatives (Reichert & Borsuk, 2005). Last, but not least, quantified uncertainty (4) facilitates learning by updating the predictions when new information arises (e.g. by monitoring changes after implementation of measures)."*

40 In addition, we made it explicit why elucidating the sources of predictive uncertainty is relevant for stakeholders (see response to comment 7 below).

**6. Section 5.1 includes the term "change signal" in the section title. What does this mean? This aspect is not actually described. At the top of page 14, it is described that "relative differences between different alternatives are often robust and can lead to a stable ranking of management alternatives." Is this what is meant by "change signal"? This point**

**would be valuable to explore and to discuss. Might some of the large uncertainty in model predictions disappear if the focus is on differences between alternatives, rather than absolute predictions of alternatives?**

We replaced 'change signal' by 'future change', because that was the true focus of that section. Regarding a stable ranking of management alternatives, we could not set up a ranking here: this being an exposure study, we had no information on the costs of alternatives and preferred endpoints of the involved stakeholders (this will be explored in a following stage).

**7. In the conclusions, I found the first sentence of the second paragraph promising: "From our results we can derive definite recommendations for practical water management." However, I felt that the recommendations that followed (e.g. uncertainty analysis should always be performed, one should follow climate change effects) were not specific enough to be practical. I would have liked to have read more about how "proper accounting of uncertainty today" will "make management fairly robust" in the next decades. By what means is such robustness actually achieved? Overall, I feel the authors need to more strongly articulate their contribution in terms of the means by which they were able to accomplish their objectives. The paper would also be more valuable if they could demonstrate how the information on uncertainty would actually be used to formulate more robust management decisions.**

Conclusions were completely re-written to provide more details about the recommendations for practical water management:

[revised manuscript text omitted]

**Review #2**

**This paper describes the application of an integrated-water-resources-management model to a small catchment in the Swiss Plateau in order to analyze potential effects of different drivers on future water quality. The authors have chosen a question as title ("Can integrative catchment management mitigate future water quality issues caused by climate change and socio-economic development?"), but I am not sure whether they really answer it. It appears that some aspects of current practice (e.g., the application of pesticides within the catchment) are so uncertain that almost no scenario leads to significant changes, whereas other factors (in particular climate change in the study area) are simply not strong enough that a model could quantify effects. This is somewhat frustrating – and I would actually have loved to get a clearer answer to the question posed in the title, which seems to be "no". Or maybe "not as long as we don't know what the farmers are actually applying on their fields".**

The conclusions were extended with an explicit answer to the question in the title:

*"The answer to the question in the title – Can integrative catchment management mitigate future water quality issues caused by climate change and socio-economic development? – from our case study is generally positive, but with some amendments. Socio-economic scenarios in this study were really influential compared to climate change for the time horizon considered (2050), yet they were compiled by local stakeholders on the ground of reality, just like the climate predictions by the creators of the ENSEMBLES database. Moreover, climate change as a whole was more uncertain than the individual model chains due to the divergence between them. The same applied to future socio-economic development: pooling all 4 scenarios together (symbolizing that we don't know which will actually happen) blurred clear individual changes into uncertainty. At the end, most management measures were powerful enough to compensate for non-manageable effects, such as climate change and socio-economic development. Therefore, by careful planning and continuous monitoring of the direction of socio-economic development and climate change, one can actually maintain the present water quality in the case study area and even improve certain components in the future."*

**I believe that the authors are stuck in a classical dilemma. They have chosen a study area with a comparably good data situation (even though the data in pesticide applications were still bad), in which not much direct climate change is expected. The temperature will rise, but won't make Northern Switzerland semi-arid. Precipitation estimates are highly uncertain and projections even don't agree on the sign of the change. Even before performing the study one could have guessed that climate change will have a smaller impact on water quality in the chosen catchment than changes in legislation regarding the use of certain pesticides. It would have been much more exciting to perform the analysis in a Mediterranean setting where much more severe climate-change effects can be expected, and maybe also more change in land use. Unfortunately, the data situation in these countries with respect to micropollutants is typically much worse than in Switzerland. Thus, while the authors' intension is a noble endeavor, their choice of application leads to some inconclusiveness.**

We rephrased the results section on climate change (4.3) to emphasize that climate change is strong but uncertain:

*"However, because only one climate change path will develop, the analysis of single model chains proves that water quality may change substantially depending on the actual future climate."*

**I was not able to understand what the hydrological model does. Also the supplementary information is illusive in this regard. In particular, it has been impossible for me to figure out how evapotranspiration is modeled, which in the study area most likely will more strongly be influenced by climate change than precipitation. The authors talk about three flow components (baseflow, subsurface flow, runoff), but it's not clear to me which physics are behind the separation**
5 **between baseflow and subsurface flow. Neither could I see which flow component is how important. Maybe this has been described in one of the two preceding papers; but I have not read them.**

We added more details about the hydrologic model to section (see response to Editor's comments). The description of the flow separation routine was extended as well (see response to comment 3 of Reviewer #1).

**The paper contains hardly any description of the catchment. However, the appropriateness of the model choice strongly**
10 **depends on the geological setting and the agricultural land-use management. The authors need to explain this. My own quick research yielded: The geology is dominated by upper freshwater molasse (poorly permeable bedrock) with limited Quaternary overburden containing peat (particularly close to lake Greifensee). There is practically no groundwater body. The Mönchaltorfer Aa is connected to several drainage channels and/or tile drains, which exist essentially all over the valley. This information may justify a model concept that essentially denies explicit groundwater pathways.**
15 **The infiltrating water is rapidly captured by the tile drains, so that restricting transformation processes to soil layers may be OK. However, other catchments are quite different and require an explicit treatment of groundwater flow, transport, and management. That is, integrated water resources management in the chosen catchment has the stream (and lake Greifensee) as its target, whereas in other catchments drinking-water production from groundwater is a major issue. In as much, "integrated water resources management" means different things in different catchments,**
20 **requiring different model concepts.**

The catchment description was extended with geological details and the importance of groundwater and drains. Please see response to Editor's comments.

**The authors choose comparably simple descriptions for all processes, which I can understand given the difficulty of calibrating more complex models. However, if the transformation behavior of the pesticides is mainly based on calibrating**
25 **simple first- order elimination models, the important influences of climate and land-use may be neglected. Some processes within the model contain an influence of air temperature (which is not identical with soil temperature). But I have not seen a potential influence of soil moisture, which may change more than precipitation in a warmer climate because of stronger evapotranspiration. Personally, I believe that including such effects will still not lead to strong climate signals in Northern-Swiss water quality. But, the best stochastic analysis does not help if the decisive dependencies are**
30 **lacking in a model. Conversely, identifying the decisive dependencies requires data that cover a sufficient range in the controlling variables. In as much, I agree with some of the statements made by the authors in their discussions regarding the uncertainty caused by not identifying the decise mechanisms. While I don't have a solution either, it's not clear to me what the authors are recommending. Should we do more stochastic analysis on parameters that we can handle even though we know that the highest model uncertainty is on conceptual levels?**

35 We agree that the stochastic analysis can't mend the model. However, a proper uncertainty analysis will translate model deficiencies into stochastic variability and reveal the true uncertainty of model predictions to the decision maker. A good practice seems to be periodically analyzing key system properties that indicate or even foretell system transitions. This was integrated into the conclusions as parts of the practical recommendations and the outline on how uncertainty analysis can improve management. Please refer to response to comment 7 of Reviewer #1.

40 **I am sorry that my (very much delayed) review contains remarks that are almost philosophical rather than going into specifics of the model and the application. But is appears that most of the latter was described in the two former papers. Therefore, it's also not easy to grasp what was actually done in this particular study without studying the preceding papers by the same authors.**

The entire water quality model presented here is novel, we emphasize this novelty more (please see responses to Reviewer #1). Following the Editor's suggestions, we included more details about the already published hydrological model into the main text.

[revised manuscript text omitted]